# COVID-19 misinformation: Mere harmless delusions or much more? A knowledge and attitude cross-sectional study among the general public residing in Jordan

Malik Sallam[1,2,3]*, Deema Dababseh[4], Alaa Yaseen[1], Ayat Al-Haidar[4], Duaa Taim[4], Huda Eid[4], Nidaa A. Ababneh[5], Faris G. Bakri[6,7,8], Azmi Mahafzah[1,2]

1 Department of Pathology, Microbiology and Forensic Medicine, School of Medicine, The University of Jordan, Amman, Jordan, 2 Department of Clinical Laboratories and Forensic Medicine, Jordan University Hospital, Amman, Jordan, 3 Department of Translational Medicine, Faculty of Medicine, Lund University, Malmö, Sweden, 4 School of Dentistry, The University of Jordan, Amman, Jordan, 5 Cell Therapy Center (CTC), The University of Jordan, Amman, Jordan, 6 Department Internal Medicine, School of Medicine, The University of Jordan, Amman, Jordan, 7 Department of Internal Medicine, Jordan University Hospital, Amman, Jordan, 8 Infectious Diseases and Vaccine Center, University of Jordan, Amman, Jordan

* malik.sallam@ju.edu.jo

**Data Availability Statement:** All relevant data are within the manuscript and its Supporting Information files.

## Abstract

Since the emergence of the recent coronavirus disease 2019 (COVID-19) and its spread as a pandemic, media was teeming with misinformation that led to psychologic, social and economic consequences among the global public. Probing knowledge and anxiety regarding this novel infectious disease is necessary to identify gaps in knowledge and sources of misinformation which can help public health efforts to design and implement more focused interventional measures. The aim of this study was to evaluate the knowledge, attitude and effects of misinformation about COVID-19 on anxiety level among the general public residing in Jordan. This cross-sectional study was conducted using an online-based questionnaire that took place in April 2020, which targeted people residing in Jordan, aged 18 and above. The questionnaire included items on the following: demographic characteristics of the participants, knowledge about COVID-19, anxiety level and misconceptions regarding the origin of the pandemic. The total number of participants included in final analysis was 3150. The study population was predominantly females (76.0%), with mean age of 31 years. The overall knowledge of COVID-19 was satisfactory. Older age, males, lower monthly income and educational levels, smoking and history of chronic disease were associated with perceiving COVID-19 as a very dangerous disease. Variables that were associated with a higher anxiety level during the pandemic included: lower monthly income and educational level, residence outside the capital (Amman) and history of smoking. Misinformation about the origin of the pandemic (being part of a conspiracy, biologic warfare and the 5G networks role) was also associated with higher anxiety levels. Social media platforms, TV and news releases were the most common sources of information about the pandemic. The study showed the potential harmful effects of misinformation on the general public and

**Funding:** The authors received no specific funding for this work.

**Competing interests:** The authors have declared that no competing interests exist.

emphasized the need to meticulously deliver timely and accurate information about the pandemic to lessen the health, social and psychological impact of the disease.

## Introduction

The entire world faced an unprecedented challenge in the form of the most recent pandemic caused by severe acute respiratory syndrome coronavirus 2 (SARS-CoV-2) [1]. Coronavirus disease 2019 (COVID-19) resulted in a massive number of infections throughout the world with a higher mortality rate among high risk groups (elderly, those with comorbidities) [2]. The public was left in a state of disarray due to the socio-economic consequences of the pandemic [3–5]. This global phenomenon dominated the media and became part of everyday conversation [6, 7]. The emergence of this virus led to a worldwide lockdown, army enforced rules, disruption of education and a shift in the global economy [4]. The swift implementation of these measures and rapid escalation in number of cases and deaths caused by the virus may have caused a state of uncertainty among the general public, which demonstrates the significance of providing correct knowledge and reliable information for proper management of this public health emergency [5, 8].

The clinical manifestations of COVID-19 vary, but commonly include: fever, cough, shortness of breath, vomiting and diarrhea [9–11]. The virus is primarily transmitted via respiratory droplets and close contact with an infected person [11]. SARS-CoV-2 can remain active for hours and even days on surfaces, therefore, touching infected surfaces can lead to the spread of infection [12, 13]. To date, there are limited therapeutic options and no vaccine available for COVID-19 infection, and management hinges on supportive therapy [14]. This is why frequent hand washing and social distancing are the ideal protective measures [15].

The World Health Organization (WHO) has declared early on during the course of COVID-19 pandemic the existence of an accompanying "infodemic" [16]. This infodemic was defined as "an over-abundance of information–some accurate and some not–that makes it hard for people to find trustworthy sources and reliable guidance when they need it" [16]. Inaccurate or false information that are communicated regardless of the deception intent is termed "misinformation" [17, 18]. This includes the circulation of conspiracy theories that prevail at times of fear and uncertainty [19]. Conspiracy theories regarding the origins of COVID-19 might be a way for the public to make sense of this pandemic. However, dangerous speculations about the virus might diminish the efforts in controlling the spread of infection [8, 20]. Thus, it is important to assess the misconceptions and misbeliefs among the public which can reveal defects that should be targeted by awareness tools [21].

The potential negative effects of COVID-19 misinformation have been the subject of active research since the onset of the pandemic [8, 22–26]. Our previous investigation of this topic entailed students at the University of Jordan (UJ) with results pointed to an association between the belief in conspiracy regarding the origin of the virus and a lower COVID-19 knowledge accompanied with higher anxiety level [27]. In Jordan, strict governmental-issued infection control measures that included wide lockdowns, curfew, mask and social distancing enforcement, and prohibition of large gatherings were helpful in delaying the first wave of COVID-19 epidemic in the country. However, these measures can be viewed currently as "delaying the inevitable", since the number of daily diagnosed cases of COVID-19 escalated rapidly from about 500 active cases in late August 2020, to reach more than 60,000 active cases by the end of October 2020 [28].

Deciphering the level of knowledge and attitude toward this unprecedented pandemic can help in identifying the current gaps in knowledge about COVID-19. Resources must be

utilized to bridge this gap and promote proper knowledge about COVID-19, which in turn will help in disease control. Studying anxiety is of prime importance as well, since it may drive the public behavior and attitude towards the infection control and mitigation measures [29, 30]. Thus, the objectives of the current study were: (1) to assess the overall COVID-19 knowledge and attitude among the general public residing in Jordan, (2) to evaluate the effects of misinformation regarding COVID-19 origins on the anxiety level, and (3) to assess the main sources of knowledge regarding the disease in the country.

## Methods

### Study design and description of the questionnaire

This cross-sectional study was conducted using an online-based questionnaire that took place between April 11, 2020 (21:00) to April 14, 2020 (00:00), thus spanning 75 hours and targeting residents in Jordan aged 18 years and above. Participation in the study was voluntary and an informed consent was included. Participants were recruited via sending mass invitations to the contacts of the authors through WhatsApp groups, and by posting public announcements on Facebook and Twitter accounts, as well as on public Facebook groups that share interests and opinions regarding the Jordanian society, asking the participants to share the survey with their contacts. The questionnaire comprised six sections with a total of 39 items addressing various subjects regarding knowledge, attitude, misinformation, sources of knowledge, and anxiety of participants regarding COVID-19. The language used to conduct the survey was Arabic (S1 Appendix).

The questionnaire contained items on socio-demographic information (age, nationality, sex, governorate of residence, marital status, monthly family income, educational level, history of smoking, and the presence of any chronic disease). Four items were used to assess the attitude towards the quarantine period (perception of the danger of the disease, adherence to quarantine measures, spending quality time with the family, and annoyance by the inability to attend religious houses of worship).

Two items were used to assess the sources of information, and three items were included to determine the role of conspiracy theories, biological warfare, and 5G networks in the origin and the spread of the pandemic. An additional item was also included to examine the belief in a divine role in the origin of the disease (S1 Appendix). Incomplete survey, manifested by item-non-response, was allowed and analysis was done using 'available-case' approach.

### COVID-19 knowledge score (K-score) calculation

Thirteen items were used to assess the overall COVID-19 knowledge among the study participants. These items included knowledge of symptoms of the disease (fever, cough, vomiting and diarrhea, and shortness of breath), knowledge of virus transmission (touching infected surfaces, close contact with an infected person, and transmission via blood), infectivity of the virus on surfaces for long periods of time, use of antibiotics for treating the disease, availability of a vaccine, remedial effect of garlic, onion and ginger on the infection, ability of summer heat to inactivate the virus, and possibility of reinfection by the virus (S1 Appendix). Individual K-scores were considered valid and included in the analyses if the participant provided responses to all 13 items.

### Assessment of the anxiety score

The final section of seven items was used to measure the level of anxiety during the government-enforced quarantine period using the General Anxiety Disorder-7 (GAD-7) scale [31].

This scale is a reliable method for anxiety assessment and included four possible responses to each item. The study participants were asked how often they have been bothered by each of the seven core symptoms of generalized anxiety disorder during the past two weeks. Response options are "not at all," "several days," "more than half the days," and "nearly every day," scored as 0, 1, 2, and 3, respectively. Individual anxiety-scores were considered valid and included in the analyses if the participant provided responses to all seven items. The maximum possible anxiety score was 21 with the minimum being zero.

## Source of information about COVID-19

To study the most common sources of COVID-19 information, we allowed the participants to select a single option out of the following choices: Ministry of Health official website, scientific journals, medical doctors, television programs and news releases, or social media platforms. If they selected social media, another single option was required to be answered (Facebook, Instagram, Twitter, or WhatsApp, S1 Appendix).

## Ethical permission

The study was approved by the Department of Pathology, Microbiology and Forensic Medicine and by the Scientific Research Committee at the School of Medicine/UJ, (using WhatsApp conference call) which was later registered under the reference number 2479/2020/67 at the School of Medicine/UJ. An informed consent was ensured by the presence of an introductory section of the questionnaire, with submission of responses implying the agreement to participate. All collected data were treated confidentially.

## Statistical analysis

All statistical analyses were conducted in IBM SPSS v22.0 for Windows. Significance was considered for P-values <0.050. We used the chi-squared ($\chi^2$) test to evaluate the significance of relationships between categorical variables. For continuous variables (e.g. age), we used the Mann-Whitney $U$ (M-W) to compare the mean among two independent groups, and the Kruskal Wallis (K-W) test to compare the mean among more than two independent groups.

# Results

## Characteristics of the study population

The total number of individuals who participated in the survey and that were included in final analysis was 3150 after filtering out responses from those who were less than 18 years old. This resulted in a minimum of 1.8% margin of error (an expression of how much the response of the study sample is representative of the general population), considering the 95% confidence interval and the current total population of Jordan (10,184,790 people), which is related to the items with response from all study participants [32, 33]. General features of the study participants are summarized in (Table 1). The median age of the study participants was 27 (mean: 31, interquartile range: 22–37). Females dominated the study population (n = 2358, 76.0% with 47 cases of non-response), and residents of the Central region of Jordan represented 84.1% of the participants (with 78 cases of non-response). For educational level and monthly income, the majority of study participants had an undergraduate degree (diploma or bachelor's degrees) representing 73.6% and the majority had a monthly income of less than 1000 JODs (n = 2413, 78.3%, Table 1).

**Table 1. Socio-demographic characteristics of the study population.**

| Feature | | Number | Percentage |
|---|---|---|---|
| **Sex** | *Male* | 745 | 24.0% |
| | *Female* | 2358 | 76.0% |
| **Nationality** | *Jordanian* | 2894 | 92.8% |
| | *Non-Jordanian* | 223 | 7.2% |
| **Region**[a] | *North* | 375 | 12.2% |
| | *Central* | 2584 | 84.1% |
| | *South* | 113 | 3.7% |
| **Marital status** | *Single* | 1617 | 51.8% |
| | *Married* | 1414 | 45.3% |
| | *Divorced* | 59 | 1.9% |
| | *Widow/widower* | 29 | 0.9% |
| **Monthly income** | *Less than 500 JOD*[c] | 1233 | 40.0% |
| | *500–1000 JOD* | 1180 | 38.3% |
| | *More than 1000 JOD* | 668 | 21.7% |
| **Educational level**[b] | *High school or less* | 496 | 15.8% |
| | *Undergraduate degree* | 2310 | 73.6% |
| | *Postgraduate degree* | 333 | 10.6% |
| **Smoking** | *non-Smoker* | 2328 | 74.4% |
| | *Smoker* | 803 | 25.6% |
| **History of chronic disease** | *No* | 2858 | 91.1% |
| | *Yes* | 278 | 8.9% |

[a]Region: North region includes the following Jordanian governorates: Irbid, Ajloun, Jerash and Mafraq; Central region includes Balqa, Amman (the capital), Zarqa and Madaba; South region includes: Karak, Tafilah, Ma'an and Aqaba.

[b]Educational level: Undergraduate degree includes diploma and bachelor's degrees; postgraduate degree includes masters and Doctor of Philosophy degrees.

[c]JOD: Jordanian Dinar.

Note: Despite having 3150 as the total number of study participants, the numbers above might not add up to reach this total, due to the existence of partial response to some survey items.

## COVID-19 knowledge

The overall knowledge of the study participants regarding COVID-19 is illustrated in (Fig 1). The majority of the study participants correctly responded to eleven out of the thirteen items, with the least percentage of correct responses observed in the following two items: the virus can remain active on surfaces for few hours (49.7%) and reinfection by COVID-19 is not possible (23.7%, Fig 1).

The total number of participants who had a valid K-score was 2988, with a mean K-score of 10.2 (range: 1.0–13.0). Higher level of knowledge regarding COVID-19 was seen among residents of Amman (mean K-score 10.3 vs. 10.1, p = 0.003; M-W), participants with higher income (10.5 vs. 10.2 vs. 10.0, p<0.001; K-W), participants with higher educational level (10.6 vs. 10.2 vs. 9.8, p<0.001; K-W) and among non-smokers (10.2 vs. 10.0, p = 0.016, M-W; Fig 2).

In addition, participants who felt annoyed by the inability to attend places of worship displayed a lower mean K-score (10.1 vs. 10.4, p<0.001; M-W). For the survey items related to COVID-19 misinformation, the participants who believed that the pandemic is related to a conspiracy and those who believed that it was part of a biologic warfare had lower mean K-score (10.0 vs. 10.4, p<0.001 for both comparisons; M-W). In addition, those who believed in the role of 5G networks in COVID-19 spread had a lower mean K-score (9.8 vs. 10.3, p<0.001; M-W). Moreover, those who believed that COVID-19 is a spiritual divine test showed a lower

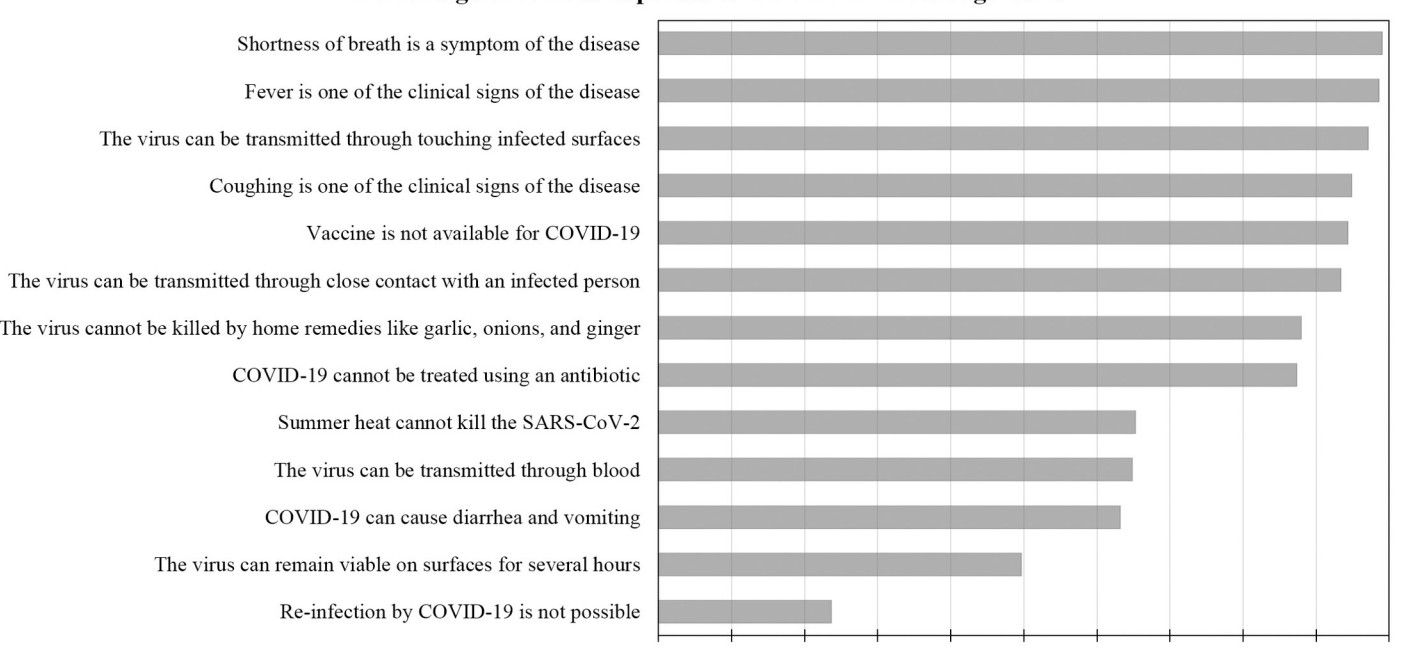

**Fig 1. The overall knowledge of COVID-19 among the study participants shown as the percentage of correct responses to each of the thirteen items assessing COVID-19 knowledge.** All items were shown as correct statements. COVID-19: coronavirus disease 2019, SARS-CoV-2: severe acute respiratory syndrome coronavirus 2.

K-score compared to those who did not hold such a belief (10.1 vs. 10.4, p<0.001; M-W, Fig 3).

## Perception of COVID-19 danger and attitude towards quarantine

The majority of study population felt that the disease is moderately dangerous (n = 1896, 60.3%), followed by 1152 participants who perceived the disease as very dangerous (36.6%). Older participants felt the disease as more dangerous compared to younger participants (40.6% vs. 31.7%), p<0.001; $\chi^2$, Table 2). Males were more likely to perceive the disease as very dangerous compared to females (41.2% vs. 35.2%, p = 0.001; $\chi^2$). Participants with the lowest monthly income were more inclined to feel that COVID-19 is very dangerous (41.0% vs. 34.5% vs. 32.5%, p = 0.002; $\chi^2$). Also, those with the lowest educational level were more likely to believe that the disease is very dangerous (44.0% vs. 34.8% vs. 38.1%, p = 0.001; $\chi^2$). Smokers (42.8% vs. 34.6%) and those with history of chronic disease (48.2% vs. 35.5%) had higher likelihood to perceive COVID-19 as a very dangerous disease (p<0.001 for both comparisons; $\chi^2$). Those who believed that the current pandemic was a spiritual test were more likely to perceive the disease as very dangerous (38.0% vs. 29.8%, p = 0.001; $\chi^2$). Married participants were more likely to feel that COVID-19 is very dangerous compared to single participants (40.6% vs. 32.5%, p<0.001; $\chi^2$, Table 2). The vast majority of study participants reported adhering to the quarantine measures (n = 3072, 97.9%). Variables that were associated with higher likelihood to break the quarantine measures included males (4.3% vs. 1.3%, p<0.001; $\chi^2$) and history of smoking (3.7% vs. 1.4%, p<0.001; $\chi^2$, S2 Appendix).

## Beliefs and misinformation about COVID-19 origin

**1. Is COVID-19 part of a global conspiracy?.** Overall, a total of 1501 of the study participants believed that COVID-19 is part of a global conspiracy (47.9%, Fig 4). This belief was

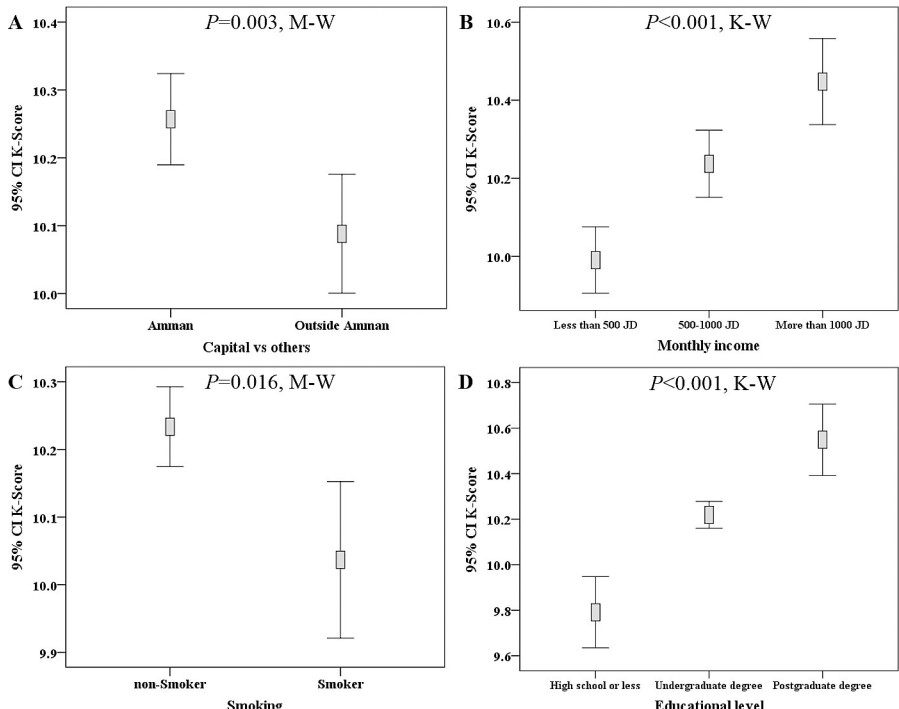

**Fig 2. Demographic features correlated with differences in COVID-19 knowledge score (K-score) among the study participants.** Higher K-score was seen among residents of Amman (A), participants with higher monthly income (B), non-smokers (C), and among participants with higher educational level (D). M-W: Mann-Whitney *U* test, K-W: Kruskal-Wallis test, CI: confidence interval of the mean K-score, JOD: Jordanian dinar.

more common among females compared to males (50.1% vs. 41.2%, p<0.001; $\chi^2$), among married participants compared to single participants (50.5% vs. 45.8%, p = 0.011; $\chi^2$) and among smokers compared to non-smokers (52.8% vs. 46.3%, p = 0.001; $\chi^2$). The gradual increase in monthly income was associated with a gradual decrease in the belief that COVID-19 is part of a global conspiracy (50.5% in those with income of <500 JOD vs. 48.2% in those with income of 500–1000 JOD vs. 43.8% among those with income of >1000 JOD, p = 0.019; $\chi^2$). For educational level, the belief in conspiracy was the highest among those with a lower educational level (50.4% among those with high school or less degree vs. 48.5% among those with an undergraduate degree, vs. 40.8% among those with postgraduate degrees, p = 0.016; $\chi^2$).

**2. Is COVID-19 part of a biological warfare?.**   The majority of study participants had a belief that SARS-CoV-2 origin was related to biological warfare (n = 1778, 57.0%, Fig 4). This belief was more common among females (59.7% vs. 48.6%, p<0.001; $\chi^2$), married participants (59.2% vs. 55.0%, p = 0.023; $\chi^2$), participants with low and middle income (p = 0.001; $\chi^2$), and lower educational level (p = 0.002; $\chi^2$).

**3. Do 5G networks have a role in COVID-19 spread?.**   The overall belief in 5G networks role in the spread of COVID-19 was generally less compared to the previously mentioned items (conspiracy and biological warfare) (n = 641, 21.0%, Fig 4). This misbelief was higher among females (23.6% vs. 12.8% in males, p<0.001; $\chi^2$), married participants (23.1% vs. 19.4% among singles, p = 0.017; $\chi^2$), lower monthly income (24.2%, vs. 20.9% vs. 15.9%, p<0.001; $\chi^2$), and lower educational level (28.0% vs. 20.4% vs. 15.5%, p<0.001; $\chi^2$).

**4. Is COVID-19 a spiritual divine test or trial?.**   The majority of study participants stated that COVID-19 pandemic is a divine spiritual test (n = 2595, 82.7%). Variables associated with

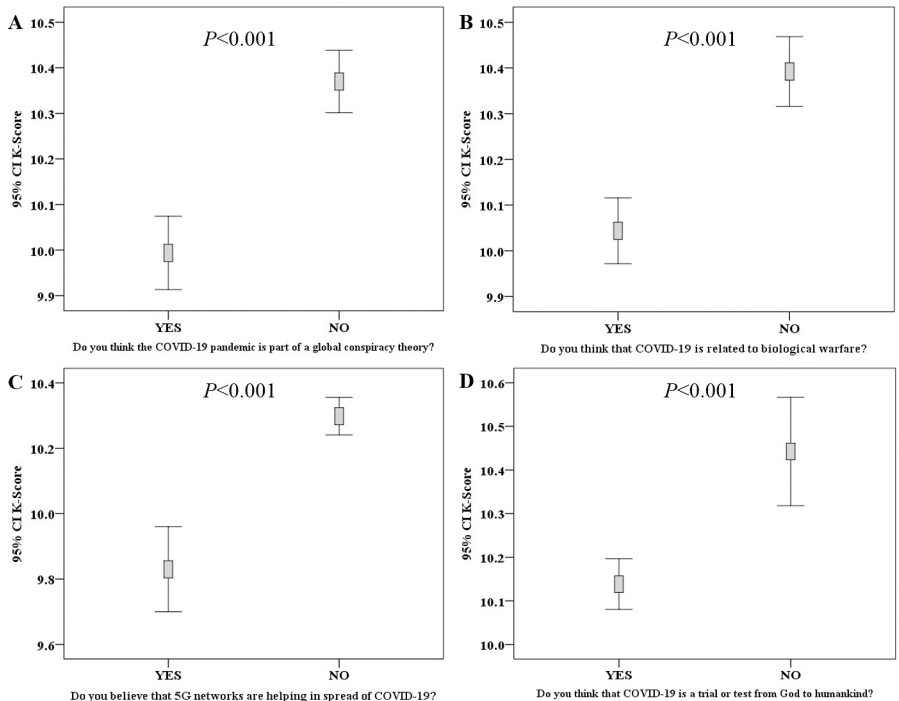

**Fig 3. Correlation between COVID-19 knowledge score (K-score) and items assessing misinformation regarding origin of COVID-19 pandemic.** Lower K-score was seen among participants who stated that COVID-19 is part of a global conspiracy plot (A), participants who stated that COVID-19 is related to biologic warfare (B), participants who believed in the role of 5G networks in COVID-19 spread (C), and participants who thought that COVID-19 is a divine test (D). P-values were calculated using Mann-Whitney *U* test, CI: confidence interval of the mean K-score.

such a belief included females (85.0% vs. 75.8%, p<0.001; $\chi^2$), Jordanian nationality (83.3% vs. 77.1% among non-Jordanians, p = 0.019; $\chi^2$), residence outside Amman (86.5% vs. 80.1% among those in Amman, p<0.001; $\chi^2$), marriage (86.5% vs. 79.3, p<0.001; $\chi^2$), lower educational level (88.2%, vs. 82.9% vs. 73.5%, p<0.001; $\chi^2$), lower monthly income (88.4% vs. 82.9% vs. 71.6%, p<0.001; $\chi^2$), and non-smoking (84.2% vs. 78.5%, p<0.001; $\chi^2$).

## Anxiety regarding COVID-19

The total number of participants who had a valid anxiety score was 3035, with mean score of 9.2 (range: zero-21.0). Variables with significant association to higher anxiety level were female sex (mean anxiety score: 9.3 vs 8.7, p = 0.007, M-W), residence outside Amman (9.5 vs. 9.0, p = 0.006; M-W), lower educational level (10.1 vs. 9.1 vs. 8.5, p<0.001; K-W), lower monthly income (9.9 vs. 9.0 vs. 8.3, p<0.001; K-W), and smoking (9.9 vs. 9.0, p<0.001; M-W, Fig 5). In addition, those who felt annoyed by the inability to attend places of worship had a higher mean anxiety score (9.7 vs. 8.2, p<0.001; M-W). The participants who thought that the quarantine gave them an opportunity to spend a quality time with their families had a lower mean anxiety score (8.9 vs. 10.8, p<0.001; M-W). Higher anxiety scores were found among those who believed that COVID-19 is related to conspiracy (9.7 vs. 8.7; p<0.001; M-W), biological warfare believers (9.6 vs. 8.6, p<0.001; M-W), those who believed in the role of 5G networks in facilitating COVID-19 spread (10.3 vs. 8.9, p<0.001; M-W), and those who believed that the pandemic is a spiritual divine test (9.4 vs. 8.0, p<0.001; M-W, Fig 6).

**Table 2. Perception of COVID-19 danger among study participants.**

| Characteristic | Self-reported COVID-19 danger | Not dangerous | Moderately dangerous | Very dangerous | P-value[c] |
|---|---|---|---|---|---|
| | | N[b] (%) | N (%) | N (%) | |
| **Age group** | Less than 27 | 36 (2.5) | 939 (65.8) | 453 (31.7) | <0.001 |
| | More than or equal to 27 | 52 (3.4) | 849 (55.9) | 617 (40.6) | |
| **Sex** | Male | 31 (4.2) | 407 (54.6) | 307 (41.2) | 0.001 |
| | Female | 65 (2.8) | 1463 (62.1) | 829 (35.2) | |
| **Marital status** | Single | 49 (3.0) | 1042 (64.4) | 526 (32.5) | <0.001 |
| | Married | 46 (3.3) | 794 (56.2) | 573 (40.6) | |
| | Divorced | 1 (1.7) | 31 (52.5) | 27 (45.8) | |
| | Widow/widower | 1 (3.4) | 12 (41.4) | 16 (55.2) | |
| **Monthly income** | Less than 500 JOD[a] | 38 (3.1) | 690 (56.0) | 505 (41.0) | 0.002 |
| | 500–1000 JOD | 37 (3.1) | 735 (62.3) | 407 (34.5) | |
| | More than 1000 JOD | 21 (3.1) | 430 (64.4) | 217 (32.5) | |
| **Educational level** | High school or less | 21 (4.2) | 257 (51.8) | 218 (44.0) | 0.001 |
| | Undergraduate degree | 67 (2.9) | 1439 (62.3) | 803 (34.8) | |
| | Postgraduate degree | 9 (2.7) | 197 (59.2) | 127 (38.1) | |
| **Smoking** | non-Smoker | 65 (2.8) | 1457 (62.6) | 805 (34.6) | <0.001 |
| | Smoker | 31 (3.9) | 428 (53.3) | 344 (42.8) | |
| **History of chronic disease** | No | 92 (3.2) | 1750 (61.3) | 1015 (35.5) | <0.001 |
| | Yes | 4 (1.4) | 140 (50.4) | 134 (48.2) | |

[a]JOD: Jordanian dinar.

[b]N: Number.

[c]P-value: Calculated using chi-squared test ($\chi^2$).

**Beliefs of the study participants regarding origin and spread of COVID-19**

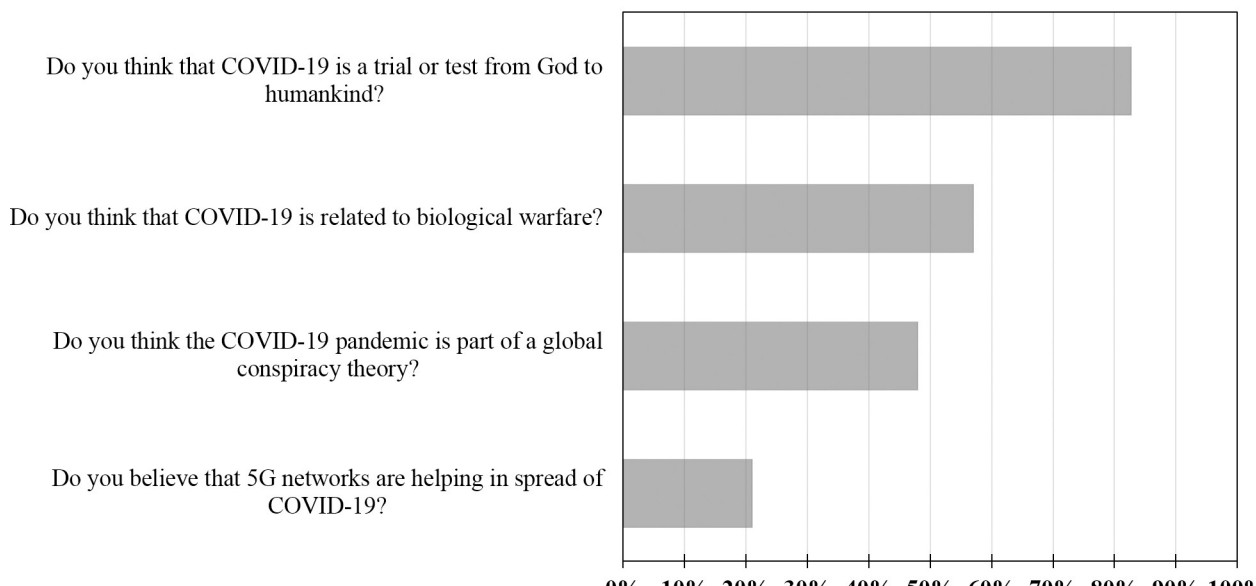

**Fig 4. The overall belief of the study participants with regard to origin and spread of COVID-19.** COVID-19: coronavirus disease 2019, 5G: the 5[th] generation mobile network.

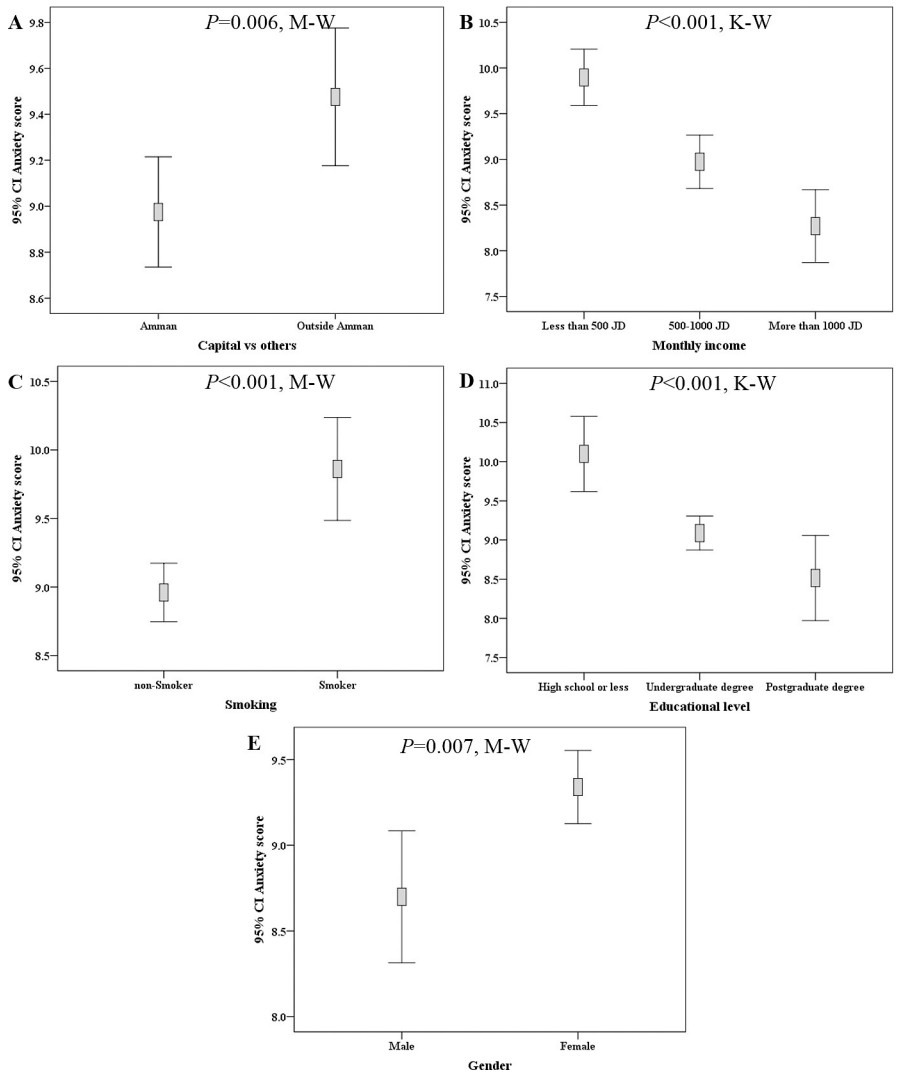

**Fig 5. Demographic features correlated with differences in anxiety score among the study participants.** Lower anxiety score was seen among residents of Amman (A), participants with a higher monthly income (B), non-smokers (C), among participants with a higher educational level (D), and among males (E). M-W: Mann-Whitney *U* test, K-W: Kruskal-Wallis test, CI: confidence interval of the mean anxiety score, JOD: Jordanian dinar.

## The main media and other sources of information about the pandemic

The most common source of information for study participants regarding the pandemic were social media platforms (n = 1075, 34.4%), followed by TV and news releases (n = 850, 27.2%), the official Ministry of Health website on COVID-19 (n = 771, 24.6%) and finally scientific journals and opinion of medical doctors (n = 432, 13.8%). For social media platforms, Facebook predominated as the main source of information about the pandemic (81.1%), followed by WhatsApp (7.0%), YouTube (4.6%), Twitter (4.5%) and Instagram (2.9%).

The participants who relied on TV and news releases as the main source of knowledge about the virus were older in age compared to those who used other sources combined (33 vs. 30 years old, p<0.001; M-W). Those who relied on doctors and scientific journals had a higher mean K-score (10.5 vs. 10.1, p<0.001; M-W) and a lower mean anxiety score (8.1 vs. 9.3, p<0.001; M-W, Fig 7).

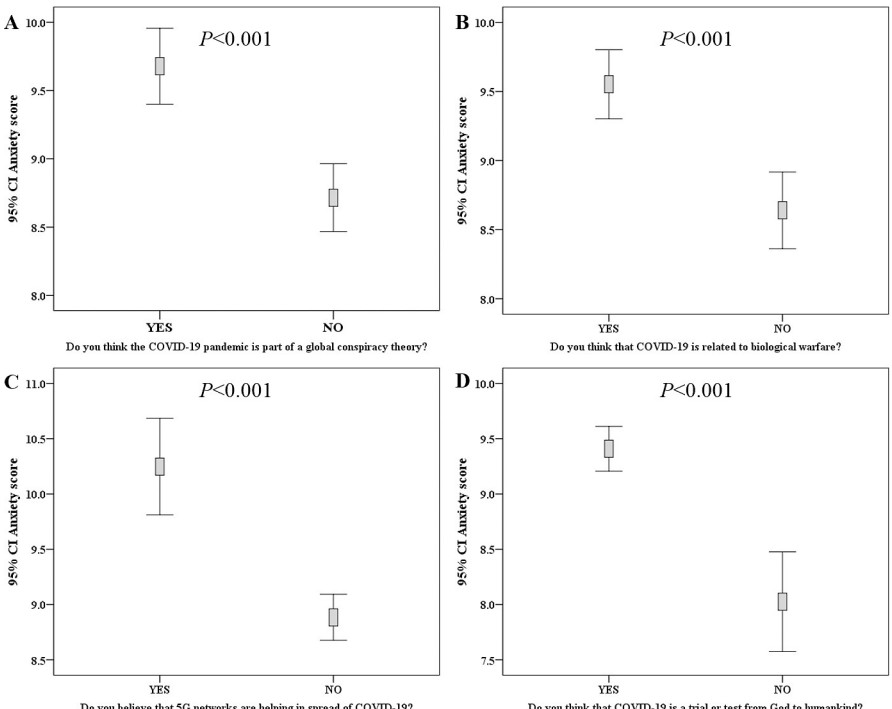

**Fig 6. Correlation between anxiety score and items assessing misinformation regarding origin of COVID-19 pandemic.** Higher anxiety score was seen among participants who stated that COVID-19 is part of a global conspiracy plot (A), participants who stated that COVID-19 is related to biologic warfare (B), participants who believed in the role of 5G networks in COVID-19 spread (C), and participants who thought that COVID-19 is a divine test (D). P-values were calculated using Mann-Whitney $U$ test, CI: confidence interval of the mean anxiety.

Participants who depended on medical doctors and scientific journals as their main source of information about COVID-19 were less predisposed to believe in conspiracy (38.4% vs. 49.3%, p<0.001; $\chi^2$), and its related misinformation (5G networks role: 14.2% vs. 22.0%, p<0.001; $\chi^2$, biological warfare role: 47.7% vs. 58.3%, p<0.001; $\chi^2$ and belief that the pandemic is a spiritual test: 69.1% vs. 48.9%, p<0.001; $\chi^2$). They were also less prone to feel that the

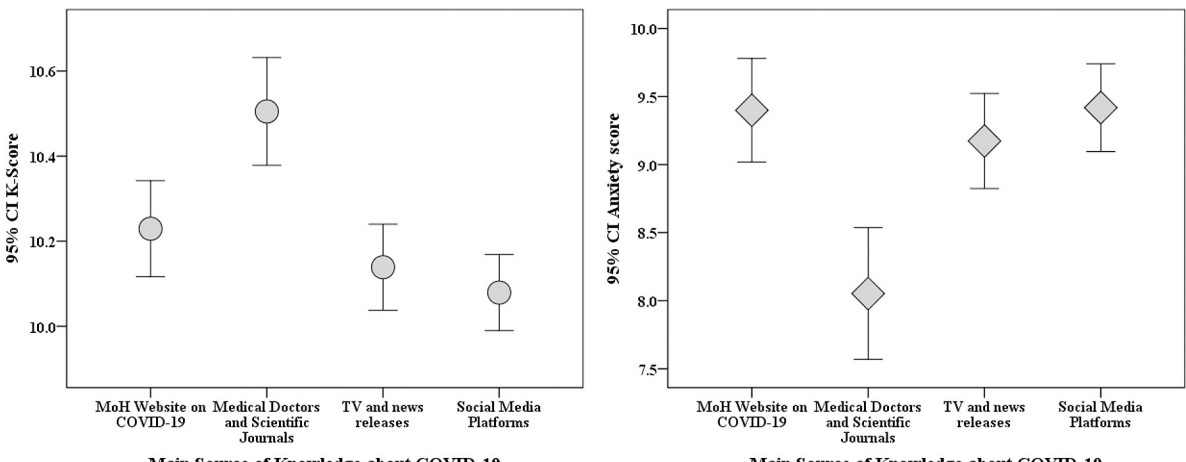

**Fig 7. Correlation between the main sources of information regarding COVID-19 and both the anxiety and K-scores.** CI: confidence interval of the mean score, K-score: COVID-19 knowledge score. MoH: Jordanian Ministry of Health.

disease is very dangerous (30.6% vs. 37.6%, p<0.001; $\chi^2$). Higher reliance on information provided by scientific journals and medical doctors was seen among males (18.8% vs. 12.4%, p<0.001; $\chi^2$), participants with a higher educational level (25.7% vs. 14.0% vs. 4.9%, p<0.001; $\chi^2$), participants with a higher monthly income (21.4% vs. 13.7% vs. 9.6%, p<0.001), residence in Amman (15.2% vs. 12.0%, p = 0.012; $\chi^2$), non-Jordanian nationality (19.9% vs. 13.4%, p = 0.007) and single marital status (17.0% vs. 10.4%, p<0.001; $\chi^2$).

## Discussion

The key results of this study can be summarized as follows: the overall knowledge of COVID-19 among residents in Jordan was satisfactory. This satisfactory knowledge was shown by the percentage of correct answers in response to the total of thirteen items that were used to assess COVID-19 knowledge in this study. Overall, more than 87% correct responses were found for eight items and more than 63% correct responses for eleven items. The participants scored less for two items: SARS-CoV-2 can remain active on surfaces for few days rather than few hours (50.3%) and re-infection by COVID-19 is not possible (23.7%). The lower knowledge in relation to these two items can be attributed to ongoing research that has not achieved a widespread outreach for the public yet. Such research indicated the stability of SARS-CoV-2 on surfaces for more than 24 hours depending on the nature of the surface [13, 34]. For the possibility of re-infection, the current evidence points to unlikely occurrence of such a phenomenon despite the need for more research tackling this aim in light of increasing reports on its genuine occurrence [35–41].

The high overall knowledge might be attributed to general interest of the public in this pandemic that became a global phenomenon and such high knowledge has been reported recently by several studies around the globe [42–47]. However, upon further dissecting COVID-19 knowledge, in relation to possible origins of the pandemic, severe gaps in knowledge were revealed. This was manifested by high prevalence of belief in conspiracy (47.9%), biologic warfare role (57.0%) and 5G networks' role (21.0%) in the origin and spread of the virus. In our previous work among university students, we found a significant association between higher anxiety during the current pandemic and the belief in conspiracy in COVID-19 origin [27]. The results of the current study showed some harmful effects of belief in conspiracy in relation to higher anxiety levels.

Even though the natural origin of SARS-CoV-2 was scientifically determined to a large extent, which further discredits the role of a conspiracy in the origin of the disease, many people still grasp to such delusions [48, 49]. The current climate of fear and uncertainty seems as a fertile soil from which conspiracy beliefs are born and thrive [50, 51]. Thus, the high prevalence of inaccurate beliefs about COVID-19 origins seems a plausible result. In addition, the link that was demonstrated between higher anxiety levels and conspiracy belief in this study, is not unique and can augment the previous evidence that such belief is harmful [27, 52, 53]. In particular, a recent study showed that belief in conspiracy is a predictor of distress and anxiety among health-care workers [54]. Another study found that such belief was associated with the presence of depression or distress [55].

The clearest variable associated with lower overall knowledge about the disease, belief in conspiracy and higher anxiety level was the lower socio-economic status (lower educational level and lower monthly income). This result is consistent with findings from various recent studies, and points to the importance of targeting such groups with intensified awareness campaigns [43, 44, 56–58].

The most common main source of information about COVID-19 reported by the participants were the social media platforms. The role of social media in fueling and spreading

implausible notions cannot be overlooked [59–61]. The spread of such misinformation via social media outlets is not a recent phenomenon that accompanied the current crisis, but also involved several health-related aspects (e.g. vaccination, AIDS denialism, Zika fever outbreak, etc.) in the last decade [61–63]. To fight against the spread of harmful misinformation, the correct public health messages should be delivered in a user-friendly style with emphasis on fact-checking tools [64]. The prime role relies on experts, physicians and the policy makers to advocate for social media campaigns that can aid in establishing a culture of fact-checking [65].

Regarding the anxiety level of the study participants and taking into account that the survey was conducted in April 2020 (early on during the course of the pandemic), the overall mean anxiety score showed a mild anxiety among the study participants. Females showed a higher anxiety level compared to males and this can be partly explained by the differences in physiology which increase females' susceptibility to develop anxiety and stress [66, 67]. Also, the lower socio-economic status was associated with a higher anxiety level which can be attributed to the lack of income security during the crisis. Moreover, participants living outside Amman had higher anxiety; this might be due to certain hardships accompanying inhabitants of rural areas and more isolated areas, like financial strains and social isolation. Such results are in line with recent research pointing to similar associations [7, 44, 68–70]. Furthermore, smokers had higher mean anxiety score compared to non-smokers; this is because of the rising emphasis on how smokers are more vulnerable to infections including COVID-19, besides the worsened prognosis in case of protracting the disease [71, 72].

For the perception of COVID-19 degree of threat, males, participants with a lower socio-economic status, smokers and those with a history of chronic disease were likely to perceive the disease as very dangerous. This result appears plausible, particularly for individuals with comorbidities, considering the high-risk of severe disease and mortality among this group [73].

Finally, one observation in this study warrants further and meticulous exploration. This entails the attitude and belief towards the origin of the disease and the government-enforced public health measures from a religious perspective. Feeling annoyed by the inability to perform religious practices due to forced closure of mosques and churches was seen in the majority of study participants (67.3%). Another observation was that 82.7% of the study population believed that the origin of the pandemic is a form of test or trial by God. A higher level of anxiety among the aforementioned groups can be attributed to the inability of a majority of participants to worship in large gatherings in either mosques or churches during the lockdown period in the country. A large sum of previous reports showed that religious belief can reduce anxiety in various health-related conditions [74–77]. Further research is needed to establish the role of religious belief in coping at time of crisis.

## Study limitations

Despite the relatively large sample size, bias was observed for sex (a majority of females) and for age. However, age seems to reflect the age distribution among the residents of Jordan at least to some extent. In addition, it was justified to have predominance of residents in the Central region as it harbours roughly two-thirds of population in Jordan including the Capital, Amman. Furthermore, we should clearly state that the results of the current study might not be representative of the Jordanian population. This is partly related to survey distribution via contacts and networks of the authors, which make sampling bias inevitable. Thus, further studies are needed to confirm our findings at different national and cultural levels. The study validity can be another limitation in relation to K-score calculation, despite having a majority of items adopted from previously published work [45]. Finally, we have to state the inherent

limitations of surveys including the response of the participants in a way they think would please the researchers, in addition to the problem of available-case analysis. However, we believe that the later analysis did not result in severe bias considering that item non-response did not exceed 1.0% for the majority of survey items (S3 Appendix).

## Conclusions

COVID-19 poses a crisis that drastically changed the world; this is illustrated by the social, psychological and economic impact of the disease. This pandemic is framed with endless streams of misinformation and fake news, which has its own consequences and spreads even more confusion. In the current research, we inspected knowledge of COVID-19 at a country level, with a special focus on the prevalence of belief in different conspiracy theories regarding the origin of the pandemic. The results of the study can provide new insights to the general public, policy makers, and media platforms to the importance of fact-checking and the potential harms of spreading misinformation, particularly those related to conspiracies surrounding the origin of this pandemic. The results of this study showed satisfactory knowledge about the disease among residents in Jordan, with large-scale lack of knowledge in certain aspects of the disease regarding origin and conspiracy surrounding this pandemic. Individuals with a lower socio-economic status showed higher anxiety, lower COVID-19 knowledge and higher belief in misinformation. Focused awareness campaigns and proper delivery of correct information is mandatory, particularly for this group to reduce the negative impact of the pandemic on their lives. Belief in the role of conspiracies, biological warfare, and 5G networks in the origin and spread of the disease was prevalent in this study and was also associated with a higher level of anxiety, which indicates that COVID-19 misinformation may have harmful effects among the general public. The spread of misinformation and conspiracies is exacerbated by different media outlets, which is why proper management and close monitoring of posted content is necessary. Our results indicated that the reliance on reliable sources to get information on the current pandemic (e.g. scientific journals and medical doctors) was associated with lower levels of anxiety. This demonstrates the significant role that should be played by the scientific community and clinicians to address the gaps in knowledge and to correct misinformation among the general public.

## Supporting information

**S1 Appendix. Consent form and questionnaire translated to English (the original form in Arabic is provided as well).**
(PDF)

**S2 Appendix. Supplementary tables (additional results).**
(PDF)

**S3 Appendix. Table on the number and percentage of item non-response in the survey.**
(PDF)

## Author Contributions

**Conceptualization:** Malik Sallam, Deema Dababseh, Azmi Mahafzah.

**Data curation:** Malik Sallam, Deema Dababseh.

**Formal analysis:** Malik Sallam.

**Investigation:** Malik Sallam, Deema Dababseh, Alaa Yaseen, Ayat Al-Haidar, Duaa Taim, Huda Eid, Nidaa A. Ababneh, Faris G. Bakri, Azmi Mahafzah.

**Methodology:** Malik Sallam, Deema Dababseh, Alaa Yaseen, Ayat Al-Haidar, Duaa Taim, Huda Eid, Nidaa A. Ababneh, Faris G. Bakri, Azmi Mahafzah.

**Project administration:** Malik Sallam.

**Supervision:** Malik Sallam, Azmi Mahafzah.

**Visualization:** Malik Sallam.

**Writing – original draft:** Malik Sallam, Deema Dababseh.

**Writing – review & editing:** Malik Sallam, Deema Dababseh, Alaa Yaseen, Ayat Al-Haidar, Duaa Taim, Huda Eid, Nidaa A. Ababneh, Faris G. Bakri, Azmi Mahafzah.

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
