## [Decision Letter · Decision Letter 0]

7 Sep 2020

PONE-D-20-21682

COVID-19 misinformation: mere harmless delusions or much more? A knowledge and attitude cross-sectional study among the general public residing in Jordan

PLOS ONE

Dear Dr. Sallam

Thank you for submitting your manuscript to PLOS ONE. After careful consideration, we feel that it has merit but does not fully meet PLOS ONE’s publication criteria as it currently stands. Therefore, we invite you to submit a revised version of the manuscript that addresses the points raised during the review process.

We look forward to receiving your revised manuscript.

Kind regards,

Flávia L. Osório, PhD

Academic Editor

PLOS ONE

Journal Requirements:

2. In your Methods section, please provide additional information about the participant recruitment method and the demographic details of your participants. Please ensure you have provided sufficient details to replicate the analyses such as: a) the recruitment date range (month and year), b) a description of any inclusion/exclusion criteria that were applied to participant recruitment, c) a table of relevant demographic details, d) a statement as to whether your sample can be considered representative of a larger population, e) a description of how participants were recruited, and f) descriptions of where participants were recruited and where the research took place.

Reviewers' comments:

Reviewer's Responses to Questions

**Comments to the Author**

1. Is the manuscript technically sound, and do the data support the conclusions?

Reviewer #1: Partly

Reviewer #2: Partly

2. Has the statistical analysis been performed appropriately and rigorously? 

Reviewer #1: No

Reviewer #2: No

3. Have the authors made all data underlying the findings in their manuscript fully available?

Reviewer #1: No

Reviewer #2: No

4. Is the manuscript presented in an intelligible fashion and written in standard English?

Reviewer #1: Yes

Reviewer #2: No

5. Review Comments to the Author

Reviewer #1: Summary: This is an interesting analysis of knowledge, attitudes and anxiety around COVID-19. The factors explored are valuable, and will add to the evidence needed on this topic. The study could be stronger with deeper data analysis, and more care around causal statements.

- Line 49: Should correctly write the full name of SARS-CoV-2 if including the full name and acronym.

- Line 54-55: this sentence seems to suggest that ‘public apprehension’ is what caused rumors to spread. Could the authors clarify?

- Lines 58-64: This paragraph discusses the clinical illness caused by SARS-CoV-2 at the beginning and end, but for some reason, the sentences in the middle discuss transmission and prevention – could the authors re-organize?

- Line 77: States that the objective of the study is to explore the ‘repercussions of misinformation’ – but this doesn’t seem to be present in lines 78-81? Except maybe ‘anxiety level’ as the sole repercussion of misinformation? I suspect there are many more multifaceted repercussions due to misinformation – would be good to clarify the scope of this study so that objectives align well.

- Methods, study design: Could the authors also include details on GAD 7 in the design section? I found detail on this later in the methods, but would be good to also include more move into the study design section.

- Overall: would be good to be consistent in terms of number of decimal places. There are times where authors use 3, 2 and 1.

- Lines 137: I am confused about what this margin of error is? Is it for responses to a specific question? Or per item in the survey? Won’t error increase when compiling an index, was that accounted for?

- Line 140: what is a non-response? Is that an incomplete survey?

- Also, were there missing data? How did the authors deal with missing responses to certain questions. Did missingness lead to bias?

- Could the authors integrate discussion on why there were so many more female respondent vs male? And how could this potentially bias/affect their findings?

- Line 334-336. While this comparison appears to hold when just comparing means, the authors should try multivariate regression to account for other key demographics factors that might be relevant to the outcomes of anxiety and belief in conspiracy. This would help address potential confounding. Without further analysis, this causal statement is a bit too strong – later in the discussion the authors point out that SES plays a role, so why wasn’t all the data utilized in the model?

- Line 358: Why does an April survey matter? Vs May or June? Did the authors expect anxiety to peak at a certain month for a specific reason?

Reviewer #2: I am pleased to have the opportunity to review this research paper. This study attempted to contribute to the knowledge of COVID-19 misinformation in Jordan. Although the research is interesting there are significant errors that should be corrected before any consideration.

First of all, the manuscript structure should be improved. I advise the following:

1. Introduction (sub-introduction parts)

2. Literature Review or Related Work. A table referring to others research studies is a good option to improve the quality of this section based on UGC.

3. Methodology. Research Hypothesis should be justified one by one according to other studies purposes.

4. Analysis of results

5 Discussion

6. Conclusion (sub-conclusions parts such as limitations and future research)

Also the abstract is incorrect. I advise the following, now the authors are focused in details that are not interesting for the reader in the abstract:

Purpose (mandatory)

Design/methodology/approach (mandatory)

Findings (mandatory)

Practical implications (if applicable)

Originality/value (mandatory)

Questions to be answered:

What practical/professional and academic consequences will this study have for the future of scientific literature about COVID-19 (theoretical contributions)? Again, the authors should make clear arguments to explain what is the originality and value of the research work. This should be stated in the final paragraphs discussion and conclusion sections.

In Conclusions part, the authors should include a final paragraph describing the “social implications" and "originality or value" of this study. What practical/professional and academic consequences will this study have in political parties strategies online?

6. PLOS authors have the option to publish the peer review history of their article (what does this mean?). If published, this will include your full peer review and any attached files.

Reviewer #1: No

Reviewer #2: No

---

## [Author Response · Author response to Decision Letter 0]

22 Sep 2020

Dear Dr. Flávia L. Osório

Regarding our manuscript with ID: PONE-D-20-21682, entitled “COVID-19 misinformation: mere harmless delusions or much more? A knowledge and attitude cross-sectional study among the general public residing in Jordan”,

Your comments and those of the reviewers were highly insightful and enabled us to improve the quality of our manuscript. In the following pages are our point-by-point responses to each of the comments of the reviewers.

Please find attached a revised version of our manuscript “COVID-19 misinformation: mere harmless delusions or much more? A knowledge and attitude cross-sectional study among the general public residing in Jordan”. The revisions were highlighted using the "Track Changes" function in the manuscript file. A clean version of the revised manuscript is submitted as well.

We hope that the revisions in the manuscript and our accompanying responses will be sufficient to make our manuscript suitable for publication in PLoS One.

Sincerely, and on behalf of the co-authors

Malik Sallam. MD, PhD

Response to comments of the Academic Editor (Dr. Flávia L. Osório)

Response: Based on the editor’s comment, we edited the style of the manuscript to meet the journal’s requirements, including the font size of headings and main text, file naming and reference style among others. Please refer to the clean version of the revised manuscript.

2. In your Methods section, please provide additional information about the participant recruitment method and the demographic details of your participants. Please ensure you have provided sufficient details to replicate the analyses such as: a) the recruitment date range (month and year), b) a description of any inclusion/exclusion criteria that were applied to participant recruitment, c) a table of relevant demographic details, d) a statement as to whether your sample can be considered representative of a larger population, e) a description of how participants were recruited, and f) descriptions of where participants were recruited and where the research took place.

Response: Based on the editor’s comment, and for the description on how the participants were recruited, we added the following statement to further clarify this issue: “Participants were recruited via sending mass invitations to the contacts of the authors through WhatsApp groups, and by posting public announcements on Facebook and Twitter accounts, as well as on public Facebook groups that share interests and opinions regarding the Jordanian society, asking the participants to share the survey with their contacts.” (page 5, lines 89-92, clean version).

Also, we believe that we have already stated most of the information required in the manuscript file. E.g. for the recruitment date, in the first paragraph of the methods section, we stated that “This cross-sectional study was conducted using an online-based questionnaire that took place between April 11, 2020 (21:00) to April 14, 2020 (00:00), thus spanning 75 hours”. (page 5, lines 85-88, clean version).

For the inclusion/exclusion criteria, the sole criterion used for exclusion was an age less than 18 years, which was stated in the methods section: “targeting residents in Jordan aged 18 years and above”. A table with detailed description of the demographic features of study participants is present in the manuscript (Table 1), which describes gender, nationality, educational level, monthly income, among other features of the study participants. (page 9, lines 165-173, clean version).

For the representation of the study sample of the larger population, the following statements are present in the discussion section, the limitations paragraph: “Despite the relatively large sample size, bias was observed for gender (a majority of females) and for age. However, age seems to reflect the age distribution among the residents of Jordan at least to some extent. In addition, it was justified to have predominance of residents in the Central region as it harbours roughly two-thirds of population in Jordan including the Capital, Amman. Furthermore, we should clearly state that the results of the current study might not be representative of the Jordanian population. This is partly related to survey distribution via contacts and networks of the authors, which make sampling bias inevitable. Thus, further studies are needed to confirm our findings at different national and cultural levels.” (Pages 20 and 21, lines 415-427, clean version).

 

Responses to the comments of Reviewer #1

Reviewer #1: Summary: This is an interesting analysis of knowledge, attitudes and anxiety around COVID-19. The factors explored are valuable, and will add to the evidence needed on this topic. The study could be stronger with deeper data analysis, and more care around causal statements.

- Line 49: Should correctly write the full name of SARS-CoV-2 if including the full name and acronym.

Response: Based on the reviewer’s comment, we re-wrote the full name of the virus as follows: “severe acute respiratory syndrome coronavirus 2”, (page 3, line 47, clean version).

- Line 54-55: this sentence seems to suggest that ‘public apprehension’ is what caused rumors to spread. Could the authors clarify?

Response: Based on the question raised by the reviewer, we rephrased this statement to make it clear as follows: “The swift implementation of these measures and rapid escalation in number of cases and deaths caused by the virus may have caused a state of uncertainty among the general public.” (Page 3, lines 53-56, clean version).

Our explanation is related to the previous paragraph that describes the measures taken by many countries in the world in response to the pandemic including the “lockdown, army enforced rules, disruption of education and a shift in the global economy”, which took place rapidly and resulted in a state of uncertainty among the general public. Also, the lack of accurate and sometimes conflicting reports regarding different aspects and future of the pandemic created a state of fear and anxiety among the general public. 

- Lines 58-64: This paragraph discusses the clinical illness caused by SARS-CoV-2 at the beginning and end, but for some reason, the sentences in the middle discuss transmission and prevention – could the authors re-organize?

Response: Based on the reviewer’s suggestion, we re-organized the paragraph as follows: “SARS-CoV-2 can remain active for hours and even days on surfaces, therefore, touching infected surfaces can lead to the spread of infection. To date, there are limited therapeutic options and no vaccine available for COVID-19 infection, and management hinges on supportive therapy. This is why frequent hand washing and social distancing are the ideal protective measures.” (Page 3, lines 59-63, clean version).

- Line 77: States that the objective of the study is to explore the ‘repercussions of misinformation’ – but this doesn’t seem to be present in lines 78-81? Except maybe ‘anxiety level’ as the sole repercussion of misinformation? I suspect there are many more multifaceted repercussions due to misinformation – would be good to clarify the scope of this study so that objectives align well.

Response: Based on the reviewer’s suggestion, we deleted the first statement of the objectives paragraph to make the section clearer, as the objectives numbered in the following sentences entail the major aims of the study, including the previous deleted statement. (Page 4, lines 76-79, clean version).

- Methods, study design: Could the authors also include details on GAD 7 in the design section? I found detail on this later in the methods, but would be good to also include more move into the study design section.

Response: Based on the reviewer’s suggestion, we added the following statement regarding the generalized anxiety disorder screener (GAD-7) in the general population. “The study participants were asked how often they have been bothered by each of the seven core symptoms of generalized anxiety disorder. Response options are “not at all,” “several days,” “more than half the days,” and “nearly every day,” scored as 0, 1, 2, and 3, respectively.” (Page 6, lines 127-130, clean version).

- Overall: would be good to be consistent in terms of number of decimal places. There are times where authors use 3, 2 and 1.

Response: Based on the reviewer’s suggestion, we re-checked the number of decimal places throughout the manuscript and we adhered to one decimal place for percentages and scores, and three places for p values.

- Lines 137: I am confused about what this margin of error is? Is it for responses to a specific question? Or per item in the survey? Won’t error increase when compiling an index, was that accounted for?

Response: The margin of error is an expression of how much the response of the study sample is representative of the general population. The smaller margin of error indicates more representation of the general population. Thus, we used the overall margin of error to assess for the representation of our study sample. However, the reviewer raised an important point, which is related to responses to specific questions. Since there are partial responses to some survey items, the margin of error may vary for each question. Thus, we added the following statement to the results section: “This resulted in a minimum of 1.8% margin of error considering the 95% confidence interval and the current total population of Jordan (10,184,790 people), which is related to the items with response from all study participants”. (Page 8, lines 145-148, clean version).

- Line 140: what is a non-response? Is that an incomplete survey?

Response: We would like to thank the reviewer for raising this important issue and enabling us to clarify it in the manuscript. We meant that some participants in the study did not provide all information, and they provided responses to some items only. Thus, we changed the non-response into “item non-response”, and we made the following changes to the methods and discussion as follows: In the methods section “Incomplete survey, manifested by item-non-response, was allowed and analysis was done using ‘available-case’ approach”. (Page 5, lines 104-105, clean version).

 In the limitations paragraph of the discussion section “Finally, we have to state the inherent limitations of surveys including the response of the participants in a way they think would please the researchers, in addition to the problem of available-case analysis. However, we believe that the later analysis did not result in severe bias considering that item non-response did not exceed 1.0% for the majority of survey items.” (Pages 20 and 21, lines 423-427, clean version).

We also believe that the issue of item non-response did not severely result in bias in our analysis as the percentage of item non-response was below 1.0% for the majority of survey items (Table below).

Table. Item non-response in the survey

Survey Item Valid responses Missing responses Item non-response Percentage

Age 2947 203 6.4

K-Score 2988 162 5.1

Anxiety score 3035 115 3.7

Do you believe that 5G networks are helping in spread of COVID-19? 3053 97 3.1

Governorates of Jordan 3072 78 2.5

Monthly income 3081 69 2.2

Gender 3103 47 1.5

Nationality 3117 33 1.0

Marital status 3119 31 1.0

Do you think that COVID-19 is related to biological warfare? 3122 28 0.9

Main Source of Knowledge about COVID-19 3128 22 0.7

Smoking 3131 19 0.6

Do you think the COVID-19 pandemic is part of a global conspiracy theory? 3133 17 0.5

History of chronic disease 3136 14 0.4

Do you think that the quarantine helped you spend a quality time with your family? 3136 14 0.4

Are you adhering to government quarantine rules and staying home? 3137 13 0.4

Educational degree 3139 11 0.3

Do you think that COVID-19 is a trial or test from God to humankind? 3139 11 0.3

Do you feel annoyed regarding your inability to practice prayers in places of worship (mosque or church)? 3144 6 0.2

Is COVID-19 a dangerous disease? 3145 5 0.2

- Also, were there missing data? How did the authors deal with missing responses to certain questions. Did missingness lead to bias?

Response: Please refer to our detailed response to the previous point above.

- Could the authors integrate discussion on why there were so many more female respondent vs male? And how could this potentially bias/affect their findings?

Response: As is was mentioned in the limitations part of the discussion section, bias was observed for gender (a majority of females). It was hard for us to find a reason for such an outcome of females being predominant in our sample, however, one potential explanation is that one step in recruitment was via approaching contacts of the authors (six out of nine were females). But we don’t feel that this was a major reason of such an outcome, as the survey announcement and invitation for participation was done using public pages and groups in social media outlets (mostly Facebook and Twitter). For the majority of analyses undertaken in this study, gender was taken into account and we noticed that females to perceive the disease as less dangerous compared to males and they tended to believe more in the role of conspiracy.

- Line 334-336. While this comparison appears to hold when just comparing means, the authors should try multivariate regression to account for other key demographics factors that might be relevant to the outcomes of anxiety and belief in conspiracy. This would help address potential confounding. Without further analysis, this causal statement is a bit too strong – later in the discussion the authors point out that SES plays a role, so why wasn’t all the data utilized in the model?

Response: We would like to thank the reviewer for raising this issue. Based on the reviewer’s comment, we conducted multinomial logistic regression analysis were gender, income, COVID-19 knowledge and anxiety remained linked to believe in the role of conspiracy in the origin of the pandemic, and added the significant results to the methods and results sections as follows:

In methods: “To analyze the variables associated with higher likelihood of believing in the role of conspiracy in the origin of the pandemic, we conducted multinomial logistic regression analysis using the following variables (age, gender, nationality, monthly income, educational level, K-score and anxiety score).” (Page 7, lines 135-138, clean version).

In results section: “Using multinomial logistic regression analysis, we found an association of belief in conspiracy regarding the origin of the pandemic with a lower K-score (p<0.001), a higher anxiety score (p=0.006), female gender (p<0.001), and lower monthly income (p=0.029).” (Page 15, lines 289-292, clean version).

Do you think the COVID-19 pandemic is part of a global conspiracy theory? (Answer: Yes) Significance Exp(B) 95% Confidence Interval for Exp(B)

Gender Male vs. female <0.001 0.704 0.584 0.849

Age Less than 27 vs. more or equal 27 0.057 0.853 0.725 1.005

Nationality Jordanian vs. Non-Jordanian 0.454 1.129 0.821 1.553

Residence Amman vs. outside Amman 0.177 1.120 0.95 1.319

Monthly income Less 500 vs. more 1000 0.029 1.278 1.025 1.593

 500 to 1000 vs. more than 1000 0.175 1.160 0.936 1.436

Educational level High school or less vs. postgraduate degree 0.172 1.257 0.906 1.745

 Undergraduate degree vs. postgraduate degree 0.066 1.285 0.983 1.680

K-score Less than 10 vs. ≥ 10 <0.001 1.551 1.299 1.851

Anxiety score Less than 9 vs. ≥ 19 0.006 0.803 0.686 0.94

- Line 358: Why does an April survey matter? Vs May or June? Did the authors expect anxiety to peak at a certain month for a specific reason?

Response: We would like to thank the reviewer for helping us to clarify this point. The response to COVID-19 in Jordan evolved over time, as the government activated the National Defense Law starting in late March that enforced lockdowns and curfew with closure of universities, schools, and prohibited public gatherings and religious practices in large masses including the closure of mosques and churches. With time, these measures started to loosen, and we believe that this might have an effect on anxiety levels regarding the pandemic. Based on the reviewer’s comment we added the following statement in the discussion section: “early on during the course of the pandemic”. (Page 19, line 389, clean version).

 

Responses to the comments of Reviewer #2

Reviewer #2: I am pleased to have the opportunity to review this research paper. This study attempted to contribute to the knowledge of COVID-19 misinformation in Jordan. Although the research is interesting there are significant errors that should be corrected before any consideration.

First of all, the manuscript structure should be improved. I advise the following:

1. Introduction (sub-introduction parts), 2. Literature Review or Related Work, A table referring to others research studies is a good option to improve the quality of this section based on UGC, 3. Methodology. Research Hypothesis should be justified one by one according to other studies purposes, 4. Analysis of results, 5 Discussion, 6. Conclusion (sub-conclusions parts such as limitations and future research)

Response: We would like to thank the reviewer for this suggestion; however, we believe that we followed the manuscript description by PLoS One (https://journals.plos.org/plosone/s/submission-guidelines#loc-manuscript-organization). Thus, we prefer to keep the manuscript style in the current format.

Also the abstract is incorrect. I advise the following, now the authors are focused in details that are not interesting for the reader in the abstract:

Purpose (mandatory)

Design/methodology/approach (mandatory)

Findings (mandatory)

Practical implications (if applicable)

Originality/value (mandatory)

Response: Based on the reviewer’s comment we made minor changes to the abstract. However, we followed the guidelines by PLoS One regarding the manuscript and abstract formatting. Also, we believe that we included all aforementioned details in the abstract. E.g. purpose: “The aim of this study was to evaluate the knowledge, attitude and effects of misinformation about COVID-19 on anxiety level among the general public residing in Jordan”, (Page 2, lines 26-28, clean version). Design: “This cross-sectional study was conducted using an online-based questionnaire that took place between April 2020” (Page 2, lines 28-29, clean version). For findings, we believe that the major results were included in the abstract. Originality/value: “The study showed the potential harmful effects of misinformation on the general public and emphasized the need to meticulously deliver timely and accurate information about the pandemic to lessen the health, social and psychological impact of the disease.” (Page 2, lines 41-43, clean version)

Questions to be answered:

What practical/professional and academic consequences will this study have for the future of scientific literature about COVID-19 (theoretical contributions)? Again, the authors should make clear arguments to explain what is the originality and value of the research work. This should be stated in the final paragraphs discussion and conclusion sections.

Response: The reviewer’s comment is very valuable and based on the second question we edited the conclusions section to accommodate these points (please refer to the conclusions section of the revised manuscript and the response to the question below).

In Conclusions part, the authors should include a final paragraph describing the “social implications" and "originality or value" of this study. What practical/professional and academic consequences will this study have in political parties strategies online?

Response: Based on the reviewer’s suggestion, we added the following paragraph to the conclusions section: “In the current research, we inspected knowledge of COVID-19 at a country level, with a special focus on the public embrace of conspiracy regarding the origin of the pandemic. The results of the study can provide new insights to the general public, policy makers, and media platforms to the importance of fact-checking and the potential harms of spreading misinformation, particularly those related to conspiracies surrounding the origin of this pandemic.” (Page 21, lines 433-437, clean version)

---

## [Decision Letter · Decision Letter 1]

16 Oct 2020

PONE-D-20-21682R1

COVID-19 misinformation: mere harmless delusions or much more? A knowledge and attitude cross-sectional study among the general public residing in Jordan

PLOS ONE

Dear Dr. Sallam,

Thank you for submitting your manuscript to PLOS ONE. After careful consideration, we feel that it has merit but does not fully meet PLOS ONE’s publication criteria as it currently stands. Therefore, we invite you to submit a revised version of the manuscript that addresses the points raised during the review process.

The article was revised by a new reviewer, who considered some changes to the current presentation of the text necessary. Several suggestions were made that will certainly improve the quality of the material presented.

We look forward to receiving your revised manuscript.

Kind regards,

Flávia L. Osório, PhD

Academic Editor

PLOS ONE

Additional Editor Comments (if provided):

Dear Authors,

Their article was revised by a new reviewer, who considered some changes to the current presentation of the text necessary. Several suggestions were made that will certainly improve the quality of the material presented.

Reviewers' comments:

Reviewer's Responses to Questions

**Comments to the Author**

1. If the authors have adequately addressed your comments raised in a previous round of review and you feel that this manuscript is now acceptable for publication, you may indicate that here to bypass the “Comments to the Author” section, enter your conflict of interest statement in the “Confidential to Editor” section, and submit your "Accept" recommendation.

Reviewer #2: All comments have been addressed

Reviewer #3: (No Response)

2. Is the manuscript technically sound, and do the data support the conclusions?

Reviewer #2: Yes

Reviewer #3: Yes

3. Has the statistical analysis been performed appropriately and rigorously? 

Reviewer #2: Yes

Reviewer #3: I Don't Know

4. Have the authors made all data underlying the findings in their manuscript fully available?

Reviewer #2: Yes

Reviewer #3: No

5. Is the manuscript presented in an intelligible fashion and written in standard English?

Reviewer #2: Yes

Reviewer #3: Yes

6. Review Comments to the Author

Reviewer #2: The authors have addressed correctly the indicated changes by this reviewer. This reviewer has not additional changes to be make to this interesting research study.

Reviewer #3: This is an interesting paper that explores knowledge, attitudes, and misinformation about COVID-19 and its effects on anxiety levels among adults in Jordan. Sources of information about the virus are also explored. The results of this paper are very important to highlight the consequences of misinformation so that the delivery of timely and accurate information is prioritized as pandemic management strategies are updated. However, the manuscript will need restructuring to better guide the reader through the paper. In particular, the authors should ensure that each analysis and writing decision always links back to the original research aims.

Below are more detailed comments and recommendations:

Major issues

Introduction

1. The introduction would benefit from supplementation and restructuring in order to provide the necessary background to frame the importance of this research, as well as to contextualize the setting in which this research was undertaken. I would suggest the following structure: (1) Global impact of COVID-19 (which the authors have already included); (2) Proper conceptualization of misinformation (e.g. definition, how it spreads) and its consequences (e.g. deaths in Iran due to false belief that ingesting methanol kills the virus); (3) Context of COVID-19 in Jordan (some details have been given in the response to reviewer 1’s last comment, but would be helpful if this were expanded upon and included in the main manuscript); (4) Why examining anxiety level is warranted/important, since it is one of the key outcome variables.

2. The objectives of the study are very clearly stated in lines 76-79 (clean version), which is great to see. These should guide the way that the subsequent sections of the paper are structured, as I will mention in a few comments below.

Methods

3. (Statistical analysis) While I see the author’s use of regression as a response to a comment from reviewer 1, it is not well justified why the most “complex” statistical approach was employed on belief in conspiracy as an outcome, since this is not one of the main research objectives. As the aim of the study is to evaluate knowledge, attitude and misinformation *on anxiety level*, if regression is used, it should be on this outcome. If the authors choose to retain their regression analyses on belief in conspiracy, why this approach was taken for this outcome in particular needs to be justified.

Results

4. While the level of detail the authors have included in their results is commendable, the sheer volume of results may be overwhelming for the reader. I suggest picking out results that will be expanded on in the discussion instead of reporting on each finding individually, and in particular, being more selective about which demographic associations to report.

5. The sections of the results should be restructured so that the research questions, as stated in the introduction, are answered in sequence: (1) Knowledge, attitudes, and misinformation; (2) The effects of (1) on anxiety; (3) COVID-19 information sources.

6. If regression analysis is used, it is important that its results are displayed in a table, and for coefficient estimates to be reported in the main text, in addition to the significance levels that are currently present in the manuscript.

Discussion

7. Discussions about the belief in conspiracy theories and anxiety levels were addressed separately, and as I understood, mainly highlighted prevalence and demographic associations, respectively. However, this discussion piece should be reworked to address effects of misinformation on anxiety in order to answer the second research objective.

Conclusion

8. The key takeaway message needs to be more clearly highlighted. Based on the title of the paper, it would seem that the main message is: not only has this paper shown that misinformation is prevalent, but that it contributes to higher anxiety levels (i.e. more than “harmless delusions).

9. Having framed the issue that this study has revealed and to round off the conclusion, the authors could consider highlighting another important finding as a solution – that knowledge from reliable information sources seems to reduce this anxiety.

Minor issues

Introduction

10. The use of the correct tense needs to be checked for consistency and/or appropriateness throughout the paper (e.g. “The entire world is facing an unprecedented challenge…” (line 46, clean version) vs. “Coronavirus disease (COVID-19) resulted in…” (line 47-48, clean version))

Methods

11. The authors may consider removing the following sentences as this information is already present elsewhere in the methods: “Participation in the study was voluntary and an informed consent was included. The questionnaire was sent through Facebook, WhatsApp, Twitter and other social media platforms.” (lines 87-89, clean version).

12. The methods section would also benefit from restructuring so that it is more intuitive for the reader to follow. The first paragraph is great, and I would recommend describing the questionnaire sections in the same sequence that they are mentioned in 93-94 (clean version): knowledge, attitude, misinformation, information sources, anxiety, and then sociodemographic variables. The descriptions of ethical permission and statistical approach can then follow.

13. In order to avoid confusion about what a “valid” K-/anxiety score may mean (e.g. that the measures were constructed from a validated scale), lines 121/122 and 129/130 (clean version) could be reworded to something like "Individual K-scores were considered valid and included in the analyses if the participant provided responses to all 13 items”.

14. It is important to describe what “as appropriate” (line 135, clean version) means – which tests were used, for what, and why?

Results

15. Similar to reviewer 1, I am also confused about the 1.8% margin of error. The response the authors provided to the reviewer would be helpful to include in the manuscript, and how this margin of error was calculated should also be described.

16. The following sentence is incomplete: “While those who thought that the quarantine gave them an opportunity to spend a quality time with their families had a lower mean anxiety score (8.9 vs. 10.8, p<0.001; M-W).” (line 221-222, clean version)

17. The following sentence needs to be revised for clarity: “Older participants perceived the disease…” (lines 243-246, clean version)

18. The authors may consider whether gender (men/women) or sex (female/male) is being examined in the study and use the appropriate language consistently throughout the paper.

19. Some results that are described are not presented within the tables or figures (e.g. spending time with family, demographic associations with conspiracy beliefs or information sources). Even if these results are not the main focus of the paper, if they are mentioned in the body of the text, they should at least be included as supplementary material.

20. The percentages reported for each source of information (from line 318, clean version), at first glance, appears quite low. From the questionnaire, I see that participants were asked to select their primary (vs. common) source of info and only allowed to select one option? This should be mentioned because it is more intuitive that people would consult more than one information source, and the phrasing “the most common source…” could be misleading.

Discussion

21. The following sentence needs to be revised for clarity: “More than 87% correct responses were found for eight items and more than 63% correct responses for eleven items out of thirteen total items that were used to assess COVID-19 knowledge in this study” (lines 349-351, clean version)

22. The authors should consider acknowledging the emergence of more recent evidence that COVID-19 reinfection may be possible (lines 356-357, clean version).

23. The sentence in lines 365-367 (clean version) should be revised. As it currently reads, it seems like the there is an association between misinformation and anxiety/knowledge as a combined item.

24. Given the length of the discussion, again, I would suggest restructuring this sentence to follow the sequence of the research objectives to make it easier to follow.

25. Could the authors expand on why they think study validity may be a limitation?

26. The table of “Item non-response in the survey” provided in the reviewer response could be added as supplementary material as evidence to support the claim made in lines 425-427 (clean version).

Conclusion

27. The phase “public embrace of conspiracy” (line 434, clean version) is a bit strong. It may be more accurate to say that the prevalence of beliefs in different conspiracy theories was investigated.

28. It seems like there may be a missing word/description after “Focused awareness” (line 441, clean version).

7. PLOS authors have the option to publish the peer review history of their article (what does this mean?). If published, this will include your full peer review and any attached files.

Reviewer #2: No

Reviewer #3: No

---

## [Author Response · Author response to Decision Letter 1]

4 Nov 2020

Dear Dr. Flávia L. Osório

Regarding our manuscript with ID: PONE-D-20-21682R1, entitled “COVID-19 misinformation: mere harmless delusions or much more? A knowledge and attitude cross-sectional study among the general public residing in Jordan”,

We would like to thank the third reviewer for the interesting, insightful and helpful comments that helped us to improve the quality and clarity of the manuscript. In the following pages are our point-by-point responses to each of the comments raised by the third reviewer.

Please find attached a revised version of our manuscript “COVID-19 misinformation: mere harmless delusions or much more? A knowledge and attitude cross-sectional study among the general public residing in Jordan”. The revisions were highlighted using the "Track Changes" function in the manuscript file. A clean version of the revised manuscript is submitted as well.

We hope that the revisions in the manuscript and our accompanying responses will be sufficient to make our manuscript suitable for publication in PLoS One.

Sincerely, and on behalf of the co-authors

Malik Sallam. MD, PhD

Responses to the comments of Reviewer #3

Reviewer #3: This is an interesting paper that explores knowledge, attitudes, and misinformation about COVID-19 and its effects on anxiety levels among adults in Jordan. Sources of information about the virus are also explored. The results of this paper are very important to highlight the consequences of misinformation so that the delivery of timely and accurate information is prioritized as pandemic management strategies are updated. However, the manuscript will need restructuring to better guide the reader through the paper. In particular, the authors should ensure that each analysis and writing decision always links back to the original research aims.

Below are more detailed comments and recommendations:

Major issues

Introduction

1. The introduction would benefit from supplementation and restructuring in order to provide the necessary background to frame the importance of this research, as well as to contextualize the setting in which this research was undertaken. I would suggest the following structure: (1) Global impact of COVID-19 (which the authors have already included); (2) Proper conceptualization of misinformation (e.g. definition, how it spreads) and its consequences (e.g. deaths in Iran due to false belief that ingesting methanol kills the virus); (3) Context of COVID-19 in Jordan (some details have been given in the response to reviewer 1’s last comment, but would be helpful if this were expanded upon and included in the main manuscript); (4) Why examining anxiety level is warranted/important, since it is one of the key outcome variables.

Response: Based on the reviewer’s suggestion, we added the following paragraph to define and show the effects of misinformation about COVID-19 at the global level as follows (in the clean version of the manuscript):

Page 3, lines 64-69: “The World Health Organization (WHO) has declared early on during the course of COVID-19 pandemic the existence of an accompanying “infodemic” [16]. This infodemic was defined as “an over-abundance of information – some accurate and some not – that makes it hard for people to find trustworthy sources and reliable guidance when they need it” [16]. Inaccurate or false information that are communicated regardless of the deception intent is termed “misinformation” [17, 18]. This includes the circulation of conspiracy theories that prevail at times of fear and uncertainty [19].”

In addition, we supplemented the paragraph on COVID-19 in Jordan to elaborate on the rapid escalation of cases since late August as follows (in the clean version of the manuscript):

Page 4, lines 78-83: “In Jordan, strict governmental-issued infection control measures that included wide lockdowns, curfew, mask and social distancing enforcement, and prohibition of large gatherings were helpful in delaying the first wave of COVID-19 epidemic in the country. However, these measures can be viewed currently as “delaying the inevitable”, since the number of daily diagnosed cases of COVID-19 escalated rapidly from late August 2020, to reach more than 60,000 active cases by the end of October 2020 [28].”

Finally, we included a paragraph to show the importance of studying the anxiety levels in time of crisis/emergency as follows (in the clean version of the manuscript):

Page 4, lines 87-88: “Studying anxiety is of prime importance as well, since it may drive the public behavior and attitude towards the infection control and mitigation measures [29, 30].”

2. The objectives of the study are very clearly stated in lines 76-79 (clean version), which is great to see. These should guide the way that the subsequent sections of the paper are structured, as I will mention in a few comments below.

Response: We would like to thank the reviewer for the comment. We will address the re-structure of subsequent sections according to the reviewer’s suggestions in the comments below (comment #12).

Methods

3. (Statistical analysis) While I see the author’s use of regression as a response to a comment from reviewer 1, it is not well justified why the most “complex” statistical approach was employed on belief in conspiracy as an outcome, since this is not one of the main research objectives. As the aim of the study is to evaluate knowledge, attitude and misinformation *on anxiety level*, if regression is used, it should be on this outcome. If the authors choose to retain their regression analyses on belief in conspiracy, why this approach was taken for this outcome in particular needs to be justified.

Response: We would like to thank the reviewer for raising this important issue that helped us to clarify our aim from conducting such analysis in response to reviewer 1. We agree that the use of belief in conspiracy as the outcome would not reflect our major aims (effect of misinformation on anxiety). In addition, we agree that the use of such a complex statistical model might not be appropriate for such an aim, in view of our previous analysis that was shown in the results section (Page 15, lines 319-323 of the clean version: “Higher anxiety scores were found among those who believed that COVID-19 is related to conspiracy (9.7 vs. 8.7; p<0.001; M-W), biological warfare believers (9.6 vs. 8.6, p<0.001; M-W), those who believed in the role of 5G networks in facilitating COVID-19 spread (10.3 vs. 8.9, p<0.001; M-W), and those who believed that the pandemic is a spiritual divine test (9.4 vs. 8.0, p<0.001; M-W, Fig 6)”. Thus, we omitted this part from the revised manuscript. However, to address the issue raised by reviewer 1, we re-phrased some strong statements in the abstract and discussion sections as follows: 

In the abstract (Page 2, lines 37-40): “Misinformation about the origin of the pandemic (being part of a conspiracy, biologic warfare and the 5G networks role) was also associated with higher anxiety levels. Social media platforms, TV and news releases were the most common sources of information about the pandemic” Instead of “Misinformation about the origin of the pandemic (being part of a conspiracy, biologic warfare and the 5G networks role) was also associated with higher anxiety and lower knowledge about the disease”.

In the discussion section (Page 18, lines 387-388): “The results of the current study showed some harmful effects of belief in conspiracy in relation to higher anxiety levels” instead of “The results of the current study clearly delineate the existence of an association between misinformation about COVID-19 and the combination of higher anxiety and lower knowledge about the disease among the public in Jordan”.

In addition, we deleted the following paragraphs (in the revised highlighted manuscript): Page 8, lines 165-168: “To analyze the variables associated with higher likelihood of believing in the role of conspiracy in the origin of the pandemic, we conducted multinomial logistic regression analysis using the following variables (age, gender, nationality, monthly income, educational level, K-score and anxiety score).” Page 16, lines 320-323: “Using multinomial logistic regression analysis, we found an association of belief in conspiracy regarding the origin of the pandemic with a lower K-score (p<0.001), a higher anxiety score (p=0.006), female gender (p<0.001), and lower monthly income (p=0.029).”

Results

4. While the level of detail the authors have included in their results is commendable, the sheer volume of results may be overwhelming for the reader. I suggest picking out results that will be expanded on in the discussion instead of reporting on each finding individually, and in particular, being more selective about which demographic associations to report.

Response: Although we agree with the reviewer regarding the length of the results section, but we still believe that we included the most significant results only. The results section might appear more succinct once the figures, figure legends and tables are formatted according the final editing by the journal. We hope that the reviewer would agree with us regarding our motivation to keep the results section in the current length after re-organization as suggested.

5. The sections of the results should be restructured so that the research questions, as stated in the introduction, are answered in sequence: (1) Knowledge, attitudes, and misinformation; (2) The effects of (1) on anxiety; (3) COVID-19 information sources.

Response: Based on the reviewer’s suggestion and to keep the flow of text in the same order based on the objectives of the study, we re-organized the results section as follows:

• Characteristics of the study population

• COVID-19 knowledge

• Perception of COVID-19 danger and attitude towards quarantine

• Beliefs and misinformation about COVID-19 origin

1. Is COVID-19 part of a global conspiracy?

2. Is COVID-19 part of a biological warfare?

3. Do 5G networks have a role in COVID-19 spread?

4. Is COVID-19 a spiritual divine test or trial?

• Anxiety regarding COVID-19

• The main media and other sources of information about the pandemic

6. If regression analysis is used, it is important that its results are displayed in a table, and for coefficient estimates to be reported in the main text, in addition to the significance levels that are currently present in the manuscript.

Response: Please refer our previous reply to comment #3.

Discussion

7. Discussions about the belief in conspiracy theories and anxiety levels were addressed separately, and as I understood, mainly highlighted prevalence and demographic associations, respectively. However, this discussion piece should be reworked to address effects of misinformation on anxiety in order to answer the second research objective.

Response: Based on the reviewer’s suggestion, we added to our previous discussion the following statements (clean version):

Pages 18-19, lines 393-399: “The current climate of fear and uncertainty seems as a fertile soil from which conspiracy beliefs are born and thrive [50, 51]. Thus, the high prevalence of inaccurate beliefs about COVID-19 origins seems a plausible result. In addition, the link that was demonstrated between higher anxiety levels and conspiracy belief in this study, is not unique and can augment the previous evidence that such belief is harmful [27, 52, 53].”

“In particular, a recent study showed that belief in conspiracy is a predictor of distress and anxiety among health-care workers [54]. Another study found that such belief was associated with the presence of depression or distress [55].”

Conclusion

8. The key takeaway message needs to be more clearly highlighted. Based on the title of the paper, it would seem that the main message is: not only has this paper shown that misinformation is prevalent, but that it contributes to higher anxiety levels (i.e. more than “harmless delusions).

Response: We would like the reviewer for helping us to strengthen and clarify the conclusions of this research. Thus, we added the following statement to the conclusions section (clean version):

Page 21, lines 468-460: “Belief in the role of conspiracies, biological warfare, and 5G networks in the origin and spread of the disease was prevalent in this study and was also associated with a higher level of anxiety, which indicates that COVID-19 misinformation may have harmful effects among the general public.” 

9. Having framed the issue that this study has revealed and to round off the conclusion, the authors could consider highlighting another important finding as a solution – that knowledge from reliable information sources seems to reduce this anxiety.

Response: We also would like to thank the reviewer for this insightful comment and based on the reviewer’s suggestion, we added the following statement to the conclusions section (clean version): 

Pages 21-22, lines 472-476: “Our results indicated that the reliance on reliable sources to get information on the current pandemic (e.g. scientific journals and medical doctors) was associated with lower levels of anxiety. This demonstrates the significant role that should be played by the scientific community and clinicians to address the gaps in knowledge and to correct misinformation among the general public.” 

Minor issues

Introduction

10. The use of the correct tense needs to be checked for consistency and/or appropriateness throughout the paper (e.g. “The entire world is facing an unprecedented challenge…” (line 46, clean version) vs. “Coronavirus disease (COVID-19) resulted in…” (line 47-48, clean version))

Response: Based on the reviewer’s comments, we made the following changes (in the clean version of the manuscript):

-Page 3, line 46: “The entire world faced an unprecedented challenge” instead of “The entire world is facing an unprecedented challenge”.

-Page 3, line 49: “The public was left in a state of disarray” instead of “The public is left in a state of disarray”

Methods

11. The authors may consider removing the following sentences as this information is already present elsewhere in the methods: “Participation in the study was voluntary and an informed consent was included. The questionnaire was sent through Facebook, WhatsApp, Twitter and other social media platforms.” (lines 87-89, clean version).

Response: Based on the reviewer’s comment, we omitted the following sentence from the study design and description of the questionnaire part of the methods section: “The questionnaire was sent through Facebook, WhatsApp, Twitter and other social media platforms” and from the ethical permission paragraph of the methods section: “Participation in the study was voluntary and anonymous.” 

12. The methods section would also benefit from restructuring so that it is more intuitive for the reader to follow. The first paragraph is great, and I would recommend describing the questionnaire sections in the same sequence that they are mentioned in 93-94 (clean version): knowledge, attitude, misinformation, information sources, anxiety, and then sociodemographic variables. The descriptions of ethical permission and statistical approach can then follow.

Response: Based on the reviewer’s suggestion, we re-organized the methods section as follows (in the clean version):

• Study design and description of the questionnaire

• COVID-19 knowledge score (K-score) calculation

• Assessment of the anxiety score

• Source of information about COVID-19

• Ethical permission

• Statistical analysis.

13. In order to avoid confusion about what a “valid” K-/anxiety score may mean (e.g. that the measures were constructed from a validated scale), lines 121/122 and 129/130 (clean version) could be reworded to something like "Individual K-scores were considered valid and included in the analyses if the participant provided responses to all 13 items”.

Response: We appreciate the kind help of the reviewer that enabled us to clarify this statement. And based on that, we changed the two sentences as follows (clean version):

Page 6, lines 126-127: “Individual K-scores were considered valid and included in the analyses if the participant provided responses to all 13 items.”

Page 6, lines 135-136: “Individual anxiety-scores were considered valid and included in the analyses if the participant provided responses to all seven items.”

14. It is important to describe what “as appropriate” (line 135, clean version) means – which tests were used, for what, and why?

Response: Based on the reviewer’s suggestion, we added the following sentences to clarify the statistical tests used in the study (in the clean version):

Page 7, lines 153-157: “We used the chi-squared (χ2) test to evaluate the significance of relationships between categorical variables. For continuous variables (e.g. age), we used the Mann-Whitney U (M-W) to compare the mean among two independent groups, and the Kruskal Wallis (K-W) test to compare the mean among more than two independent groups.”

Results

15. Similar to reviewer 1, I am also confused about the 1.8% margin of error. The response the authors provided to the reviewer would be helpful to include in the manuscript, and how this margin of error was calculated should also be described.

Response: Based on the reviewers’ comment, we added the following statement to the results section (clean version):

Page 8, lines 163-165: “This resulted in a minimum of 1.8% margin of error (an expression of how much the response of the study sample is representative of the general population)”.

The calculation was based on the online website (CheckMarket: https://www.checkmarket.com/sample-size-calculator/?fbclid=IwAR0DZS_iZnJpXTv76LXR4T5BFqE6qVGjViAtP2zy-uQAtiJ1gRBWeaEk-XQ), which has already been cited in our original submission: Reference No. 32 in the clean version: CheckMarket (2020) Calculate sample size margin of error.

16. The following sentence is incomplete: “While those who thought that the quarantine gave them an opportunity to spend a quality time with their families had a lower mean anxiety score (8.9 vs. 10.8, p<0.001; M-W).” (line 221-222, clean version)

Response: We would like to thank the reviewer for correcting this error. The sentence was corrected into (in the clean version):

Page 15, lines 316-317: “The participants who thought that the quarantine gave them an opportunity to spend a quality time with their families had a lower mean anxiety score (8.9 vs. 10.8, p<0.001; M-W).”

17. The following sentence needs to be revised for clarity: “Older participants perceived the disease…” (lines 243-246, clean version)

Response: Based on the reviewer’s suggestion, we revised the statement as follows (clean version):

Page 11, lines 236-238: “Older participants felt the disease as more dangerous compared to younger participants (40.6% vs. 31.7%), p<0.001; χ2, Table 2).”

18. The authors may consider whether gender (men/women) or sex (female/male) is being examined in the study and use the appropriate language consistently throughout the paper.

Response: Based on the reviewer’s suggestion, we used sex (female/male) instead of gender (men/women) throughout the manuscript as follows (clean version): Page 5, line 107; Page 9, Table 1; Page 13, Table 2; Page 15, line 311; Page 20, line 436.

19. Some results that are described are not presented within the tables or figures (e.g. spending time with family, demographic associations with conspiracy beliefs or information sources). Even if these results are not the main focus of the paper, if they are mentioned in the body of the text, they should at least be included as supplementary material.

Response: Based on the reviewer’s suggestion, we added a supplementary file (S2 File: Supplementary Tables [Additional Results]). This file included two tables on the attitude of the study participants to quarantine measures stratified by socio-demographic variables and on the Belief in conspiracy in relation to socio-demographic variables among the study participants. 

20. The percentages reported for each source of information (from line 318, clean version), at first glance, appears quite low. From the questionnaire, I see that participants were asked to select their primary (vs. common) source of info and only allowed to select one option? This should be mentioned because it is more intuitive that people would consult more than one information source, and the phrasing “the most common source…” could be misleading.

Response: We would like the reviewer for this comment that allowed us to clarify this issue. The questionnaire was an electronic online survey and for the main source of information, we allowed one choice to be selected. We added this part to the methods section (clean version)

Page 6, lines 138-143:

“Source of information about COVID-19

To study the most common sources of COVID-19 information, we allowed the participants to select a single option out of the following choices: Ministry of Health official website, scientific journals, medical doctors, television programs and news releases, or social media platforms. If they selected social media, another single option was required to be answered (Facebook, Instagram, Twitter, or WhatsApp, S1 Appendix).”

Discussion

21. The following sentence needs to be revised for clarity: “More than 87% correct responses were found for eight items and more than 63% correct responses for eleven items out of thirteen total items that were used to assess COVID-19 knowledge in this study” (lines 349-351, clean version)

Response: Based on the reviewer’s comment and to help increasing the clarity of the statement, we re-phrased the paragraph as follows (clean version): 

Page 18, lines 367-370: “This satisfactory knowledge was shown by the percentage of correct answers in response to the total of thirteen items that were used to assess COVID-19 knowledge in this study. More than 87% correct responses were found for eight items and more than 63% correct responses for eleven items.”

22. The authors should consider acknowledging the emergence of more recent evidence that COVID-19 reinfection may be possible (lines 356-357, clean version).

Response: Based on the reviewer’s comment, we added the following statement and a couple of references to clarify this issue (clean version):

Page 18, lines 375-377: “For the possibility of re-infection, the current evidence points to unlikely occurrence of such a phenomenon despite the need for more research tackling this aim in light of increasing reports on its genuine occurrence [35-41].”

23. The sentence in lines 365-367 (clean version) should be revised. As it currently reads, it seems like the there is an association between misinformation and anxiety/knowledge as a combined item.

Response: Based on the reviewer’s comment and in light of reviewer 1 previous comments, we edited the sentence as follows (clean version):

Page 18, lines 385-386: “The results of the current study showed some harmful effects of belief in conspiracy in relation to lower knowledge about the disease and higher anxiety levels.”

24. Given the length of the discussion, again, I would suggest restructuring this sentence to follow the sequence of the research objectives to make it easier to follow.

Response: We would like to thank the reviewer for the suggestion, but we prefer to keep the discussion in the current format and hopefully the reviewer would agree on that after reading the clean version of the manuscript.

25. Could the authors expand on why they think study validity may be a limitation?

Response: We included this statement to clarify that minor modification of some COVID-19 knowledge items should be taken into consideration upon evaluating the overall knowledge about the disease. Thus, to clarify this issue, we added the following sentence in the limitations section (clean version):

Pages 20-21, lines 443-444: “The study validity can be another limitation in relation to K-score calculation, despite having a majority of items adopted from previously published work [45]”.

26. The table of “Item non-response in the survey” provided in the reviewer response could be added as supplementary material as evidence to support the claim made in lines 425-427 (clean version).

Response: Based on the reviewer’s suggestion, we added the following supplementary file (S3 Appendix) entitled “Table on the number and percentage of item non-response in the survey.”

Clean version, Page 21, line 448: “However, we believe that the later analysis did not result in severe bias considering that item non-response did not exceed 1.0% for the majority of survey items (S3 Appendix).”

Conclusion

27. The phase “public embrace of conspiracy” (line 434, clean version) is a bit strong. It may be more accurate to say that the prevalence of beliefs in different conspiracy theories was investigated.

Response: We would like to thank the reviewer for the comment and based on that, we changed the sentence as follows (clean version):

Page 21, lines 453-455: “In the current research, we inspected knowledge of COVID-19 at a country level, with a special focus on the prevalence of belief in different conspiracy theories regarding the origin of the pandemic.”

28. It seems like there may be a missing word/description after “Focused awareness” (line 441, clean version).

Response: Based on the reviewer’s comment, we added the word “campaigns” to the sentence as follows (clean version):

Page 21, line 462: “Focused awareness campaigns and proper delivery of correct information is mandatory, particularly for this group to reduce the negative impact of the pandemic on their lives.”

Finally, we are deeply grateful for the thorough, insightful and comprehensive review made by the reviewers that helped us greatly in refining our research.

---

## [Decision Letter · Decision Letter 2]

19 Nov 2020

COVID-19 misinformation: mere harmless delusions or much more? A knowledge and attitude cross-sectional study among the general public residing in Jordan

PONE-D-20-21682R2

Dear Dr. Sallam

We’re pleased to inform you that your manuscript has been judged scientifically suitable for publication and will be formally accepted for publication once it meets all outstanding technical requirements.

Kind regards,

Flávia L. Osório, PhD

Academic Editor

PLOS ONE

Additional Editor Comments (optional):

the reviewers consider that the article is suitable for publication, once all comments and suggestions have been met. However, reviewer 2 also made two small considerations, which should be addressed by the authors.

Reviewers' comments:

Reviewer's Responses to Questions

**Comments to the Author**

1. If the authors have adequately addressed your comments raised in a previous round of review and you feel that this manuscript is now acceptable for publication, you may indicate that here to bypass the “Comments to the Author” section, enter your conflict of interest statement in the “Confidential to Editor” section, and submit your "Accept" recommendation.

Reviewer #2: All comments have been addressed

Reviewer #3: (No Response)

2. Is the manuscript technically sound, and do the data support the conclusions?

Reviewer #2: Yes

Reviewer #3: Yes

3. Has the statistical analysis been performed appropriately and rigorously? 

Reviewer #2: Yes

Reviewer #3: Yes

4. Have the authors made all data underlying the findings in their manuscript fully available?

Reviewer #2: Yes

Reviewer #3: No

5. Is the manuscript presented in an intelligible fashion and written in standard English?

Reviewer #2: Yes

Reviewer #3: Yes

6. Review Comments to the Author

Reviewer #2: The authors have addressed correctly the indicated changes. The authors have completed correctly Reviewers comments and the manuscript have improved considerately.

Reviewer #3: Thank you for addressing each of the previous comments so thoroughly - the manuscript has been substantially improved and I really enjoyed reading it.

I only have two final comments regarding the following sentence (lines 80-83, clean version):

"However, these measures can be viewed currently as “delaying the inevitable”, since the number of daily diagnosed cases of COVID-19 escalated rapidly from late August 2020, to reach more than 60,000 active cases by the end of October 2020."

1. To serve as a comparison and show the "escalation" of cases, what was the number in late August 2020?

2. Do these numbers refer to the "number of daily diagnosed cases" or "number of active cases"? This should be consistent between the two time points.

Great work!

7. PLOS authors have the option to publish the peer review history of their article (what does this mean?). If published, this will include your full peer review and any attached files.

Reviewer #2: No

Reviewer #3: No

---

## [Editor Report · Acceptance letter]

24 Nov 2020

PONE-D-20-21682R2 

COVID-19 misinformation: mere harmless delusions or much more? A knowledge and attitude cross-sectional study among the general public residing in Jordan 

Dear Dr. Sallam:

I'm pleased to inform you that your manuscript has been deemed suitable for publication in PLOS ONE. Congratulations! Your manuscript is now with our production department. 

Kind regards, 

on behalf of

Dr. Flávia L. Osório 

Academic Editor

PLOS ONE